# Biological functions of the autophagy-related proteins Atg4 and Atg8 in *Cryptococcus neoformans*

**Thiago Nunes Roberto[1], Ricardo Ferreira Lima[2], Renata Castiglioni Pascon[1], Alexander Idnurm[3], Marcelo Afonso Vallim[1] ***

**1** Departamento de Ciências Biológicas, Universidade Federal de São Paulo, Diadema, São Paulo, Brazil,
**2** Departamento de Infectologia, Universidade Federal de São Paulo, São Paulo, São Paulo, Brazil, **3** School of BioSciences, University of Melbourne, Melbourne, Victoria, Australia

* marcelo.vallim@gmail.com

**Data Availability Statement:** All relevant data are within the manuscript and its Supporting Information files.

## Abstract

Autophagy is a mechanism responsible for intracellular degradation and recycling of macro-molecules and organelles, essential for cell survival in adverse conditions. More than 40 autophagy-related (*ATG*) genes have been identified and characterized in fungi, among them *ATG4* and *ATG8*. *ATG4* encodes a cysteine protease (Atg4) that plays an important role in autophagy by initially processing Atg8 at its C-terminus region. Atg8 is a ubiquitin-like protein essential for the synthesis of the double-layer membrane that constitutes the autophagosome vesicle, responsible for delivering the cargo from the cytoplasm to the vacuole lumen. The contributions of Atg-related proteins in the pathogenic yeast in the genus *Cryptococcus* remain to be explored, to elucidate the molecular basis of the autophagy pathway. In this context, we aimed to investigate the role of autophagy-related proteins 4 and 8 (Atg4 and Atg8) during autophagy induction and their contribution with non-autophagic events in *C. neoformans*. We found that Atg4 and Atg8 are conserved proteins and that they interact physically with each other. *ATG* gene deletions resulted in cells sensitive to nitrogen starvation. *ATG4* gene disruption affects Atg8 degradation and its translocation to the vacuole lumen, after autophagy induction. Both *atg4* and *atg8* mutants are more resistant to oxidative stress, have an impaired growth in the presence of the cell wall-perturbing agent Congo Red, and are sensitive to the proteasome inhibitor bortezomib (BTZ). By that, we conclude that in *C. neoformans* the autophagy-related proteins Atg4 and Atg8 play an important role in the autophagy pathway; which are required for autophagy regulation, maintenance of amino acid levels and cell adaptation to stressful conditions.

## Introduction

Cryptococcosis is a systemic mycosis caused by encapsulated basidiomycete yeasts of the genus *Cryptococcus* [1]. The highest incidence of the disease is caused by *C. neoformans*, a global opportunistic pathogen prevalent in immunodeficient patients, mainly HIV-positive

**Funding:** This work was supported by Fundação de Amparo à Pesquisa do Estado de São Paulo (FAPESP) grants #2015/04400-9, #2016/50185-5 to MAV and #2016/14542-8 to RCP, by the SPRINT-University of Melbourne travel grant to AI, and by Coordenação de Aperfeiçoamento de Pessoal de Nível Superior (CAPES) grants #001 and #88881.133481/2016-01 to TNR.

**Competing interests:** The authors have declared that no competing interests exist.

individuals [2]. The therapy against cryptococcosis requires a long period of treatment with antifungal drugs used singly or in combination, some of which are highly toxic to the patient [3]. Due to the long period of clinical treatment, some reports indicate that fungi of the genus *Cryptococcus* may have a high potential to antifungal resistance, which could explain the therapeutic failures and recurrent relapses in the patients with cryptococcosis [4]. Thus, further studies for new strategies that contribute with the knowledge about the treatment of this mycosis, leading to the deficiency of growth, multiplication, and/or survival of the fungus in the host are of great importance. In this context, the elucidation of autophagic pathways in fungi may provide new insights into the relationship established during the infection process between pathogen and host [5].

Autophagy is an intracellular mechanism responsible for degradation and recycling of macromolecules and organelles. It is conserved in eukaryotic organisms and is essential for biological processes such as homeostasis, aging, differentiation, development, and pathogenesis [6,7]. Defects in the autophagy pathway can cause cell death and the manipulation of specific components of this system, occurring exclusively in pathogenic microorganisms but not in higher eukaryotes, may lead to a new therapeutic strategy [8]. Autophagy is a conserved process responsible for cell survival against adverse conditions such as nutritional deprivation, hypoxia or changes in carbon source [9,10]. This mechanism can be divided in three main types: chaperone-mediated autophagy, microautophagy, and selective and non-selective macroautophagy. In addition, in the model yeast *Saccharomyces cerevisiae* there is an exclusive process called Cvt (Cytoplasm to vacuole targeting), characterized as a specific type of selective macroautophagy [8,11]. Macroautophagy, hereafter referred as autophagy, is the best characterized type, responsible for the sequestration of cytoplasmic contents, involving a double-membrane vesicle termed the autophagosome, that subsequently fuses with the vacuole to promote the degradation and recycling of the transported cargo [12–14].

More than 40 autophagy-related (*ATG*) genes have been identified in fungi, however, some of them occur exclusively in a specific type of autophagy [15]. Among the central proteins involved in the autophagy pathway in *S. cerevisiae*, Atg8 is an essential ubiquitin-like protein that plays an important role in the formation of the double-layer membrane that constitutes the autophagosome vesicle [6,16]. The synthesis of the double-layer membrane depends on a set of autophagy-related proteins, especially Atg8 that needs to be in the conjugated form with the lipid phosphatidylethanolamine (PE), in order to perform its function [17,18]. In *S. cerevisiae* the conjugation of Atg8-PE is initiated by Atg4 that cleaves the arginine (residue 117) located in the C-terminal portion, exposing a conserved glycine residue. After that, Atg8 with the exposed glycine interacts with Atg7 and is then transferred to Atg3 that finally promotes the conjugation with PE [19,20]. In a second mode of action, Atg4 mediates the delipidation between Atg8 and PE, releasing Atg8 present in the external membrane of the autophagosome, thereby promoting the recycling of Atg8 in the system [21].

Recently, the autophagy mechanism involving autophagosome formation has been extended from yeast model *S. cerevisiae*, showing the need to emphasis the differences in autophagy machinery in different species. In the filamentous fungus *Aspergillus oryzae*, Atg8 is an essential protein for autophagy, and it accumulates in the vacuole after starvation or rapamycin induction. *ATG8* gene deletion results in impaired aerial hyphal growth, conidia formation and germination [22]. In *Sordaria macrospora*, a coprophilic filamentous ascomycete, Atg4 processes Atg8 in the C-terminal region, ensuring its localization in the autophagosomal vesicle. Moreover, the gene disruption leads to impaired vegetative growth and ascospore germination [23]. For the plant-pathogenic fungus *Botrytis cinerea* Atg4 plays a crucial role in autophagy by processing Atg8 [24], which is important for the formation of autophagic bodies under starvation condition [25]. Besides, it was observed that both autophagy-related proteins

are required for vegetative development, conidiation, and virulence in plant tissues [24,25]. In the rice blast fungus *Magnaporthe oryzae*, Atg4 interacts and processes Atg8, leading to accumulation of autophagic bodies in the vacuole by nitrogen starvation. In addition, Atg4 is fundamental for hyphal development, conidiation, appressorium formation, and pathogenesis [26]. In *Ustilago maydis*, a plant pathogenic fungus, Atg8 impacts on the accumulation of autophagic bodies in the vacuole, cell survival under carbon starvation condition, and teliospores formation [27]. Atg8 is also important for virulence and sporulation in the chestnut blight fungus, *Cryphonectria parasitica*, and may be a target for a mycovirus that causes a reduction in pathogenicity [28].

For the human pathogenic fungus *C. neoformans*, a variety of studies were carried out to characterize the genes associated with the autophagy pathway. Hu et al. [29] described the importance of gene *VPS34* in the autophagy process and, using RNAi, knocked down the Atg8 expression and demonstrated that this protein is important for virulence in a mouse model of cryptococcosis. Oliveira et al. [30] showed that *atg7* mutants have reduced autophagic body formation and decreased cell viability under nitrogen deprivation. Beyond defects in the autophagy pathway, the *ATG7* gene in *C. neoformans* contributes to cellular size and survival in mice lungs. Gontijo et al. [31] reported that the predicted cargo protein aspartyl aminopeptidase-like (Ape4) is important for survival at 37˚C, cell size, antifungal susceptibility, capsule formation, survival within macrophages, and virulence in mice infected with *C. neoformans*. Also, these authors pointed out that of 33 genes involved in *S. cerevisiae* autophagy, *C. neoformans* bares in its genome only 21 of them, indicating that likely the autophagy in the latter may function slightly different from what has been described for the former. Furthermore, Ding et al. [32] found that *C. neoformans ATG* genes *ATG1*, *ATG7*, *ATG8* and *ATG9* are required to maintain amino acid levels in a nitrogen depleted condition, contribute to sensitivity to the proteasome inhibitor BTZ and affect the virulence in a murine model of infection. Hence, it has been established for *Cryptococcus* the importance of Atg-related proteins in different biological aspects, yet the molecular mechanisms involving the autophagy regulation have not been fully explored. Thus, the present study aimed to investigate the *C. neoformans* Atg4 and Atg8 proteins by analyzing their complementarity of function with homologous proteins of *S. cerevisiae*, role in the regulation of autophagy pathway, and impact of mutant strains in response to multi-stress conditions.

## Materials and methods

### Fungal strains and culture conditions

The yeast strains used in this study are presented in S1 Table. The mutant strains were generated in the *C. neoformans* var. *grubii* (KN99α) and *S. cerevisiae* (BY4741) backgrounds. Cultures were commonly maintained on rich medium YPD (1% yeast extract, 2% peptone and 2% glucose). Synthetic dextrose (SD) medium was prepared either with or without ammonium sulfate and amino acids (Sigma). In order to investigate the direct protein-protein interactions by the yeast two-hybrid system, DDO (SD/+N, -Leu -Trp), QDO (SD/+N, -Leu -Trp -Ade -His), QDO/X (QDO supplemented with 40 μg/mL X-α-Gal) and QDO/X/A (QDO/X with 200 ng/mL Aureobasidin A) media were used, as indicated by the manufacturer (Clontech). Cultures were incubated at 30˚C or 37˚C, in a rotary shaker with 150 rpm when required.

### *In silico* analysis of autophagy-related proteins Atg4 and Atg8

The *C. neoformans ATG4* (CNAG_02662) and *ATG8* (CNAG_00816) genes were identified by BLASTp similarity search against the *C. neoformans* H99 database (taxid: 235443) available at the NCBI (National Center for Biotechnology Information) and using the amino acid

sequences of *S. cerevisiae* Atg4 and Atg8 as a query (SGD accession numbers YNL223W and YBL078C, respectively). The protein conserved domains and signatures were obtained using the CDD (Conserved Domain Database) platform available online at NCBI. The illustration of the protein domains was made using the IBS 1.0.3 software (CUCKOO Workgroup). Multiple alignment of homologous amino acid sequences was performed using the ClustalW program [33]. Phylogenetic analysis was conducted using the Neighbor-Joining method with 1,000 bootstrap replicates [34], using the software MEGA v. 10.0.5 [35].

## Protein-protein interactions by yeast two-hybrid assay

The direct protein-protein interactions were performed using the Matchmaker® Gold Yeast Two-Hybrid System (Clontech, Takara Bio), following the manufacturer's recommendations. For all the experiments the strain Y2HGold and the vectors pGBKT7 (bait) and pGADT7 (prey), supplied with the kit, were used. The coding sequence of each gene of interest was amplified by PCR using the specific primers listed in S2 Table. *C. neoformans ATG8* (CNAG_00816) cDNA was cloned into GAL4 DNA-binding domain vector pGBKT7, while *ATG3* (CNAG_06892); *ATG4* (CNAG_02662) and *ATG7* (CNAG_04538) cDNAs were cloned into the GAL4 transcriptional activation domain vector pGADT7. Then, the cloned vectors were individually transformed into the strain Y2HGold, according to the Yeastmaker™ Yeast Transformation System 2 (Clontech, Takara Bio) protocol, selected on SD/-Leu or SD/-Trp, and tested for the autoactivation. The activation of the reporter genes was evaluated by plating the cells on selective medium supplemented with 40 μg/mL X-α-Gal and 200 ng/mL Aureobasidin A. After autoactivation analysis, each pair of bait (pGADT7) and prey (pGBKT7) was co-transformed in Y2HGold and the transformants were selected on DDO medium. In addition, two independent clones were used to verify the interaction between bait and prey. Colonies growing in DDO were streaked onto QDO, QDO/X and QDO/X/A plates. The growth was observed after 5 days of incubation at 30˚C. The positive interactions were detected by cell survival in selective media that activate the reporter genes (*AUR1-C*, *ADE2*, *HIS3*, and *MEL1*) after bait and prey interaction. For this experiment, the plasmids pGBKT7-53 and pGADT7-T were used as a positive control, and pGBKT7-Lam and pGADT7-T were used as a negative control.

## Yeast complementation and starvation challenge

For the yeast complementation assay we amplified by PCR the coding sequence of *ATG4* (CNAG_02662) and *ATG8* (CNAG_00816) *C. neoformans* genes, using the primers indicated in S2 Table. The coding sequence fragments were cloned into the *Hind*III and *Xba*I restriction sites of the yeast expression vector pYES2 (Invitrogen) using the Gibson Assembly® Master Mix (New England BioLabs). The recombinant plasmids were introduced into the *S. cerevisiae* (BY4741) *atg4Δ* and *atg8Δ* null mutants, whose construction is described below, following the lithium acetate (LiAc)/single-stranded DNA (SS-DNA)/polyethylene glycol (PEG) transformation protocol [36]. The transformants were selected by uracil prototrophy. The functional complementation effect was tested under nitrogen starvation as proposed by Matsuura et al. [37] with some modifications. First, the cells were cultured in YPD broth for 2 days at 30˚C with 150 rpm. Then yeast cells were washed with sterile saline 0.9%, transferred to liquid SD/-N/-AA (2% glucose and 0.17% yeast nitrogen base with neither amino acids nor ammonium sulfate) and incubated for a period of 20 days at 30˚C with 150 rpm in a rotary shaker. Lastly, the culture suspension had the $OD_{530}$ adjusted to 0.6 and the suspension had the concentration reduced in a 10-fold serial dilution. Five microliters of each serial dilution were spotted on YPD plates and incubated for 48 hours at 30˚C, before comparing the growth ratio between

the wild type, mutant and/or complemented strains. The starvation challenge for *C. neoformans* was also performed using the nitrogen starvation protocol, in order to check the starvation-sensitive phenotype in the autophagy-related mutant strains. All the experiments independently were repeated three times.

### Gene deletion and fungal transformation

The *S. cerevisiae atg4Δ* and *atg8Δ* mutants were generated by substituting the *ATG4* (YNL223W) and *ATG8* (YBL078C) coding regions with the KanMX6 selectable marker, which confers resistance to G418 (Geneticin). The gene replacement was performed by a homologous recombination strategy, using the primers listed in S2 Table. The selective marker was overlap-PCR amplified from plasmid pFA6-S65TGFP-KanMX6. The PCR fragments were introduced into *S. cerevisiae* BY4741 strain using the LiAc/SS carrier DNA/PEG heat shock transformation methodology [36]. The transformants were selected on YPD plates containing 200 μg/mL G418 and the deletion was confirmed by diagnostic PCR. The *C. neoformans* gene deletions were constructed by replacing, via homologous recombination, the *ATG4* (CNAG_02662) and *ATG8* (CNAG_00816) coding regions by a selectable marker hygromycin phosphotransferase. Sequences of all the oligonucleotides primers designed for gene-disruption are indicated in S2 Table. Briefly, the selectable marker was amplified from vector pPZP-Hyg2 (hygromycin resistance cassette) (donated by Dr. J Heitman [38]) by overlap-PCR extension, based on double-joint PCR methodology [39]. Whereas, the flanking regions of a target gene to be deleted were PCR amplified using the KN99α gDNA as template. The generated fragments were introduced in wild type strain (KN99α) by biolistic transformation using the Biolistic® PDS-1000/He Particle Delivery System (BioRad) [40]. Subsequently, the mutants were plated and selected on YPD medium supplemented with 200 μg/mL hygromycin. After the gene deletion, reconstituted strains were generated by PCR amplification of the wild type native locus, that was biolistically delivered back into the mutant, co-transformed with the plasmid pPZP-Neo1 (G418 resistance cassette) (donated by Dr. J Heitman [38]). Reconstituted transformants were selected on YPD with 200 μg/mL G418. The gene replacements were tested by diagnostic PCR and confirmed by Southern blot analysis. The yeast genomic DNA extraction was performed following a previously described methodology [41] and the Southern blot technique was performed according to the protocol proposed by Sambrook et al. [42].

### GFP fusion cassette and fluorescence microscopy

The *C. neoformans ATG8* coding region was amplified by PCR and inserted into the *Bam*HI and *Spe*I cloning sites of the vector pCN50 [43], resulting in a plasmid with the gene fused in frame to the C-terminus of GFP (green fluorescent protein). The primers to generate the fragments are presented in S2 Table. The cloned vectors were introduced into wild type KN99α and *atg4* mutant strains by biolistic transformation. The selection of transformants was carried out in YPD agar containing 200 μg/mL G418 (Geneticin). The cellular localization of autophagy-related protein Atg8 was evaluated by fluorescence microscopy. In general, the yeast cells expressing the GFP fused proteins were grown overnight in liquid YPD at 30˚C with 150 rpm. The cells were harvested by centrifugation and washed twice in a phosphate-buffered saline (PBS). Then, cells were inoculated to a final concentration of $OD_{600}$ 1.0 in one milliliter of YPD and SD/-N/-AA. Each condition was incubated for 4 hours at 30˚C and 37˚C in a rotary shaker. For the vacuole membrane staining, 1 μg/mL of FM4-64 dye (Invitrogen) was added to the culture after 3 h of incubation [44]. The samples were analyzed using an Olympus BX51 fluorescent microscope with an Olympus DP73 digital camera (Olympus). A total of one hundred cells per treatment distributed in different visual fields were evaluated. All acquired

images were captured with 100× objective lens and processed with the program cellSens Dimension 1.16 (Olympus). Microscopy analyses were done in biological triplicate. The data were statistically analyzed in GraphPad Prism 7.0 software using two-way ANOVA, to compare different groups with more than one variable.

## Protein extraction and western blotting

For the total protein extract preparation, yeast cells were first cultured in YPD broth until they reached the mid-log phase at 30˚C. Autophagy was then induced by transferring the cells to liquid SD without ammonium sulfate and amino acids. Nitrogen starvation induction was performed from 0–4 hours at 30˚C and 150 rpm. After induction, the yeast cells were harvested by centrifugation and resuspended in 500 μL lysis buffer (50 mM Tris pH 7.4, 100 mM NaCl, 0.5 mM PMSF, 10 mM orthovanadate, 50 mM NaF, 0.5% NP-40) containing protease and phosphatase inhibitors (Sigma-Aldrich). Glass beads were added to the cellular suspension and cells were lysed using a Mini-Beadbeater-16 (BioSpec). The protein concentration was determined by Bradford assay [45]. For western blotting, 50 μg of total protein were separated by SDS-PAGE on 10% polyacrylamide gels and transferred to nitrocellulose membranes (Bio-Rad) using the Trans-Blot® SD Semi-Dry Transfer Cell (Bio-Rad) at 15 V for 1 hour. Then, the membranes were blocked in 5% skimmed milk in Tris-buffered saline (TBS) containing 0.1% Tween 20. Membranes were further incubated with primary mouse anti-GFP (Thermo Fisher Scientific) or rabbit anti-histone (Cell Signaling) in a 1:2,000 dilution. Following this, the membranes were incubated with anti-mouse IgG or anti-rabbit IgG (Cell Signaling) secondary antibodies conjugated to horse-radish peroxidase in a 1:5,000 dilution. The immunoreactive bands were detected using SuperSignal™ West Pico Chemiluminescent Substrate (Thermo Fisher Scientific) and visualized using ImageQuant LAS (GE Healthcare). This experiment was carried out three times independently.

## Stress sensitivity assay

The procedure to evaluate yeast growth under stress conditions was performed by incubating overnight the *C. neoformans* strains in 5 mL YPD broth at 30˚C in a rotary shaker. The cells were precipitated by centrifugation, washed twice in sterile saline and diluted to a concentration of $2 \times 10^6$ CFU/mL and 10-fold diluted until to $2 \times 10^2$ CFU/mL. Five microliters of each dilution were spotted on plates containing the stress-inducing agents. For the stress analysis, YPD agar was supplemented with 0.5%-0.75% Congo Red (cell wall integrity), 0.03%-0.04% SDS (cell membrane integrity), pH adjusted from 6 to 8 (alkaline stress), 0.75–1 M KCl (osmotic stress), 0.75–1 M NaCl (saline stress), 2.5–4 mM $H_2O_2$ (oxidative stress), and 1–2.5 mM $NaNO_2$ (nitrosative stress); furthermore, SD agar was supplemented with different combinations of ammonium sulfate and amino acids (starvation stress). The spotted cells were incubated at 30˚C and 37˚C for a period of 2 days and then photographed. Each condition was repeated in duplicate.

## Sensitivity to the proteasome inhibitor Bortezomib (BTZ)

The growth inhibition effect of Bortezomib (BTZ) upon *C. neoformans* strains (wild type and mutants) was tested as previously described by Ding et al. [32]. Briefly, an isolated colony was inoculated in 5 mL YPD and incubated overnight at 30˚C with 150 rpm. Cells were harvested by centrifugation, washed three times in a phosphate-buffered saline (PBS) and diluted to a final concentration of $10^6$ cells/mL in liquid SD/-N/-AA, supplemented or not with 50 μg/mL BTZ (Accord/Intas Pharmaceuticals). The culture was incubated for a period of 48 hours at 30˚C in a rotary shaker. To determine the fold change of the initial CFU, cells were plated at 0

and 48 h on YPD and incubated at 30˚C for 48 hours. The CFU was calculated and the data were statistically analyzed in GraphPad Prism 7.0 software using unpaired *t* test for comparison of two groups. The experiment was performed in two independent biological replicates.

## Results

### Identification of *ATG8* in *C. neoformans*

We initiated our study by identifying the *ATG8* orthologue gene in the genome of *C. neoformans*. The amino acid sequence of *S. cerevisiae* Atg8 protein (SGD accession number: YBL078C) was used as a query for the BLASTp similarity search against the *Cryptococcus neoformans* H99 database (taxid: 235443) available at the NCBI. The result revealed a putative microtubule binding protein containing 126 amino acids with 78% identity to the *S. cerevisiae* Atg8 protein, encoded in *C. neoformans* by the CNAG_00816 gene on chromosome 1 and confirming the identification previously published by other authors [29,31]. Next, we performed the phylogenetic analysis of multiple alignment using eight protein sequences with homology to Atg8 belonging to basidiomycetes and ascomycetes species. We verified that the Atg8 protein of *C. neoformans* is grouped in the same cluster of *C. gattii* and, moreover, *C. neoformans* Atg8 is phylogenetically closer to other ascomycetes species such as *Neurospora crassa*, *Aspergillus flavus* and *Magnaporthe oryzae* than *S. cerevisiae* (S1 Fig). Nevertheless, when comparing the structural sequence of different Atg8 homologous proteins between those same fungal species (S2 Fig), we verified the presence of some conserved domains (S2 Fig). The data using the CDD platform (Conserved Domain Database, NCBI) showed the presence of the Ubiquitin-like superfamily (UBQ) domain and predicted Atg4 and Atg7 proteins binding sites (filled and unfilled arrows, respectively). In addition, it was possible to verify the presence of the conserved glycine residue, necessary for the cleavage of the C-terminal portion by the cysteine protease Atg4, allowing the subsequent conjugation of Atg8 with PE [13] (S2 Fig). These results indicate that *C. neoformans* Atg8 is highly conserved with orthologues from different fungal species.

### Interactions of Atg8 with autophagy-related proteins Atg3, Atg4 and Atg7

It is known that in *S. cerevisiae* Atg8 binds directly to other autophagy-related proteins such as Atg3, Atg4 and Atg7, which are responsible for the processing of Atg8 to a form required for autophagosome vesicle synthesis [46,47]. Since *C. neoformans* Atg8 has predicted binding sites for Atg4 and Atg7 (S2 Fig), we investigated whether in *C. neoformans* there would also be direct interaction of Atg8 with these proteins, including Atg3. Direct protein-protein interactions were performed by yeast two-hybrid analysis. *C. neoformans* full-length cDNA sequences were used to construct both bait (pGBKT7) and prey (pGADT7) plasmids with the Matchmaker™ Gold Yeast Two-Hybrid System (Clontech). The results of the two-hybrid experiments performed with two independent clones demonstrated that, individually, bait and preys do not autoactivate the reporter genes (S3 Fig); however, when co-transformed, the bait pGBKT7-*ATG8* interacted with all prey containing *ATG3* (CNAG_06892), *ATG4* (CNAG_02662) or *ATG7* (CNAG_04538) sequences in all selective media (Fig 1). These observations suggest that Atg3, Atg4 and Atg7 are proteins that, by interacting with Atg8, may be involved in the autophagy pathway in *C. neoformans*.

### Characterization of autophagy-related protein Atg4 in *C. neoformans*

Since the Atg8 protein in *C. neoformans* shows the glycine residue (Atg8$^{G117}$) in the C-terminus portion (S2 Fig) and interacts directly with Atg4 via two-hybrid system (Fig 1), we decided

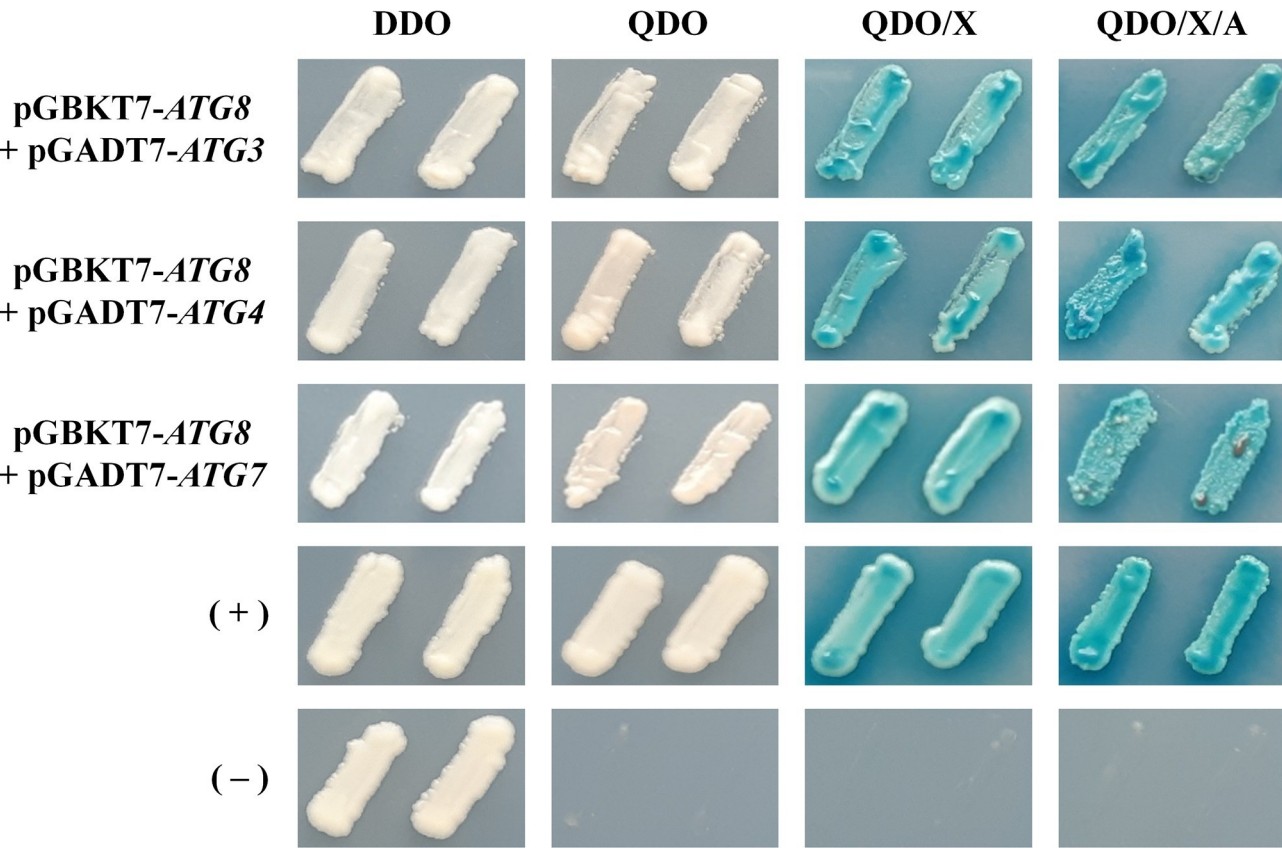

**Fig 1. Protein-protein interaction using the two-hybrid assay.** Atg8 protein of *C. neoformans* interacts with Atg3, Atg4 and Atg7. Growth of Y2HGold yeast strain co-transformed with the indicated bait (pGBKT7) and prey (pGADT7). Two independent clones were tested in a total of n = 2 independent experiments. Plates incubated for 5 days at 30°C. Assay performed with the Matchmaker™ Gold Yeast Two-Hybrid System (Clontech). DDO: SD/-Leu/-Trp; QDO: SD/-Ade/-His/-Leu/-Trp; X: 40 μg/mL X-α-Gal; A: 200 ng/mL Aureobasidin A. Pair of plasmids pGBKT7-53 and pGADT7-T were used as positive control; and pGBKT7-Lam and pGADT7-T were used as negative controls.

to include in our study the analysis of Atg4. In order to identify the *ATG4* orthologous gene, we used the amino acid sequence of *S. cerevisiae* Atg4 (SGD: YNL223W) as a query. The search in the BLASTp algorithm against the genome database of *C. neoformans* H99 (taxid: 235443) revealed similarity with the gene CNAG_02662 located in chromosome 3 and with 33% identity with the Atg4 of *S. cerevisiae*. In *C. neoformans*, *ATG4* encodes a putative cysteine protease with 1,185 amino acids. As observed in *S. cerevisiae*, Atg4 from *C. neoformans* has a conserved domain of the Peptidase_C54 Superfamily between the amino acids 480–982: this domain is present in a group of proteolytic proteins capable of hydrolyzing the peptide bond. Moreover, when aligning the multiple sequences of Atg4 homologous proteins, we observed the catalytic residue of cysteine, which is conserved in all fungi species analyzed (S4 Fig). These results suggest that the functions of Atg4 in *Saccharomyces* and *Cryptococcus* may be conserved.

### *ATG4* and *ATG8* of *C. neoformans* functionally complement the *atg4* and *atg8* mutants of *S. cerevisiae*

As we observed that Atg proteins Atg8 and Atg4 are highly conserved in different fungal species, we next performed an assay to verify the hypothesis that the autophagy-related proteins Atg4 and Atg8 of *Cryptococcus* have the same functions as their well-studied counterparts in

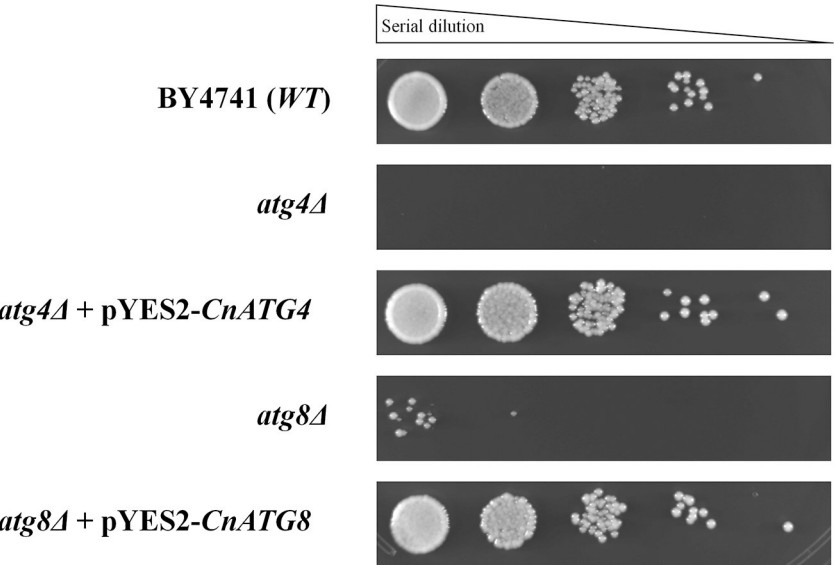

**Fig 2. Complementation of the *S. cerevisiae atg4Δ* and *atg8Δ* mutants by the respective *C. neoformans* genes.**
Growth analysis after 20 days of incubation in SD/-N/-AA broth at 30˚C and 150 rpm. Following nutritional
deprivation, serial dilutions of the cell suspension were spotted on YPD agar. Cell viability was checked after 48 hours
of incubation at 30˚C; n = 3 independent experiments. BY4741 strain was used as a growth control. The cDNA
sequences of the *ATG4* and *ATG8* genes of *C. neoformans* were cloned into the vector pYES2 (Invitrogen), *CnATG4*
and *CnATG8* respectively.

*Saccharomyces*. For this study, heterologous complementation in *S. cerevisiae* mutants was
tested. First, we obtained the *atg4Δ* and *atg8Δ* null mutants by homologous recombination in
*S. cerevisiae* (S5 Fig). Then, the full-length cDNA sequences of *C. neoformans* Atg4 and Atg8
were cloned into the expression vector pYES2 (Invitrogen) and transformed, respectively, into
the *atg4Δ* and *atg8Δ* mutants of *S. cerevisiae*. After 20 days of culture in SD/-N/-AA broth
medium, it was possible to observe a nutritional deprivation sensitivity phenotype for the *S.
cerevisiae atg4Δ* and *atg8Δ* mutants. However, when these mutants were complemented with
the *Cryptococcus* proteins, we observed that *CnATG4* and *CnATG8* recovered their growth
defects, restoring the cell survival on nitrogen-lacking medium to the wild type strain level
(Fig 2). Therefore, these results indicate that Atg4 and Atg8 of *C. neoformans* are functionally
homologous with the *S. cerevisiae* autophagy-related proteins.

## Involvement of proteins Atg4 and Atg8 in the autophagy pathway

In *S. cerevisiae* the *ATG* gene deletions leads to mutants with sensitivity to nitrogen starvation
[48]. Based on that, we decided to investigate the response to starvation for *ATG4* and *ATG8*
disrupted genes in *C. neoformans*. For this study, we first obtained individuals *atg4* and *atg8*
mutants using the double-joint methodology (S6A and S6C Fig). After confirming the gene
deletion by Southern blot analyses (S6B and S6D Fig), we reconstituted the *ATG8* gene into
the *atg8* mutant by PCR amplification of the CNAG_00816 gene, which was co-transformed
with the plasmid pPZP-Neo1 (G418 resistance). For the *atg4* mutants we tested two indepen-
dent strains, since we did not manage to restitute the wild type copy back into the mutant. For
the starvation sensitivity experiment, we induced the autophagy in the wild type (KN99α),
reconstituted (*atg8+ATG8*) and *atg* mutant strains by culturing the cells for 20 days in liquid

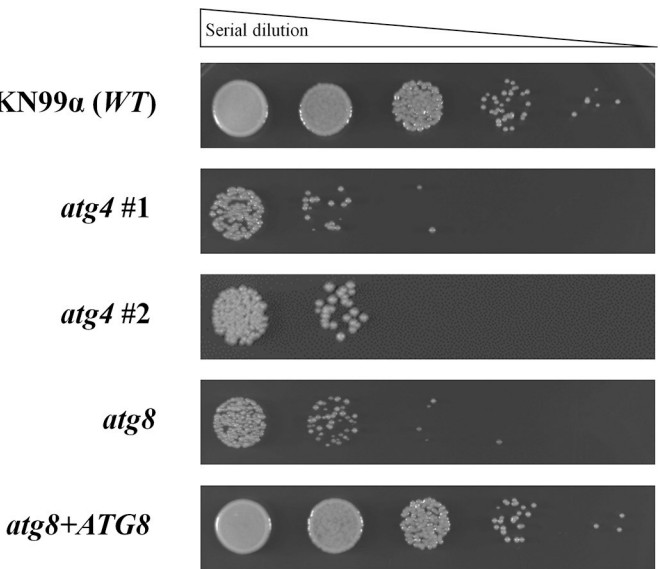

Serial dilution

KN99α (*WT*)

*atg4* #1

*atg4* #2

*atg8*

*atg8+ATG8*

**Fig 3. Starvation challenge for *C. neoformans atg4* and *atg8* mutants.** Growth sensitivity analysis after 20 days of incubation in SD/-N/-AA broth at 30˚C and 150 rpm. Following nutritional deprivation, serial dilutions of the cell suspension were spotted on YPD agar. Cell viability was checked after 48 hours of incubation at 30˚C; n = 3 independent experiments. KN99α wild type was used as a growth control.

SD/-N/-AA. After this period of starvation challenge, we observed the loss of cell viability for *atg4* and *atg8* mutants, compared to wild type and reconstituted phenotypes (Fig 3).

### Atg4 processes Atg8 during nitrogen starvation

In order to elucidate in *C. neoformans* the processing of Atg8 involving the cysteine protease Atg4 upon autophagy induction, as described in *S. cerevisiae* [19], we introduced into KN99α and *atg4* mutant strains the GFP-Atg8 plasmid. We next examined by western blot the vacuolar proteolysis of Atg8 under nitrogen starvation from a period of 0–4 h. When the wild type strain expressing GFP-Atg8 was incubated in SD/-N/-AA broth it was possible to visualize over time the bands of the GFP-Atg8 fused protein as well as the free GFP bands (Fig 4). The KN99α data suggest a normal autophagic flux, with Atg8 being degraded by releasing the GFP fragment in the course of time. On the other hand, in the *atg4* mutant only the full length GFP-Atg8 bands were detected (Fig 4), suggesting that the autophagosome synthesis was blocked and, consequently, Atg8 was not degraded.

### Atg8 intravacuolar localization occurs in an Atg4-dependent manner

In *S. cerevisiae*, the *ATG4* gene knockout generates mutant cells defective in delivering Atg8 to the vacuole during the autophagy pathway, resulting in GFP-Atg8 diffuse distribution in the cytoplasm and, consequently, indicating that Atg8 processing by Atg4 is necessary for its subsequent vacuolar association [49]. To further understand the impact of *ATG4* deletion in the mechanism of autophagy in *C. neoformans*, we next evaluated the Atg8 cellular localization by fluorescence microscopy, using the strains containing the GFP fused proteins. For that, we monitored the GFP-Atg8 localization upon cell cultivation in rich medium (YPD) and starvation medium (SD/-N/-AA) for 4 hours at 30˚C and 37˚C, after labeling the vacuole membrane with FM4-64 (Fig 5). In KN99α, the fluorescence visualization of cells growing in rich medium

## A

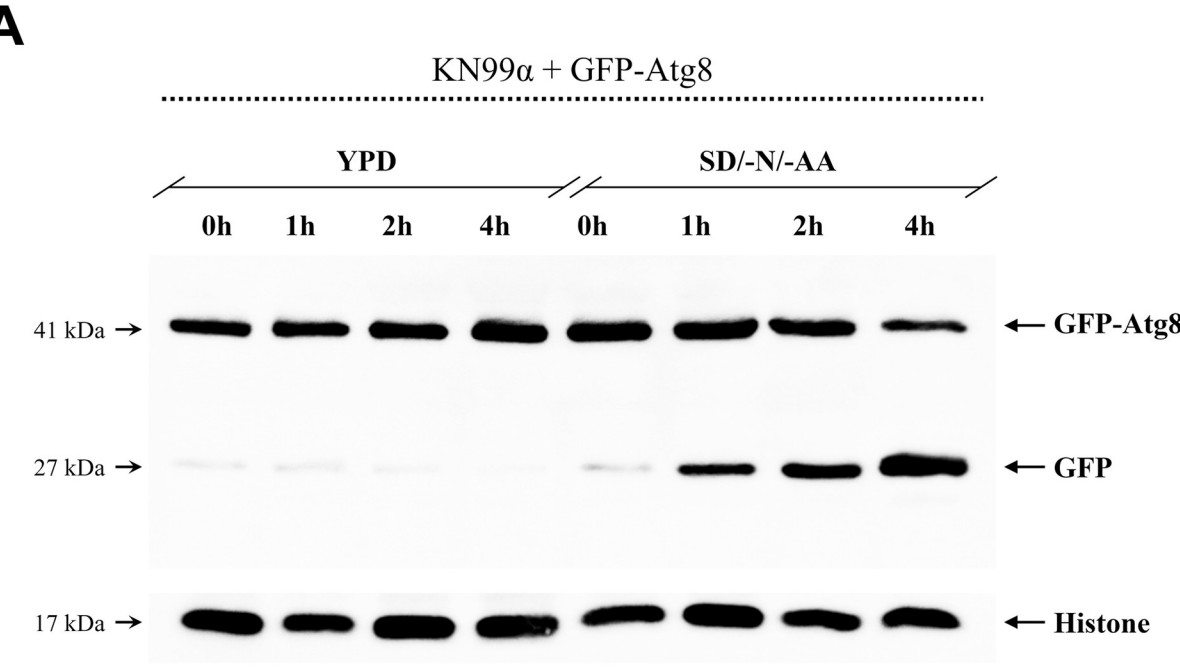

## B

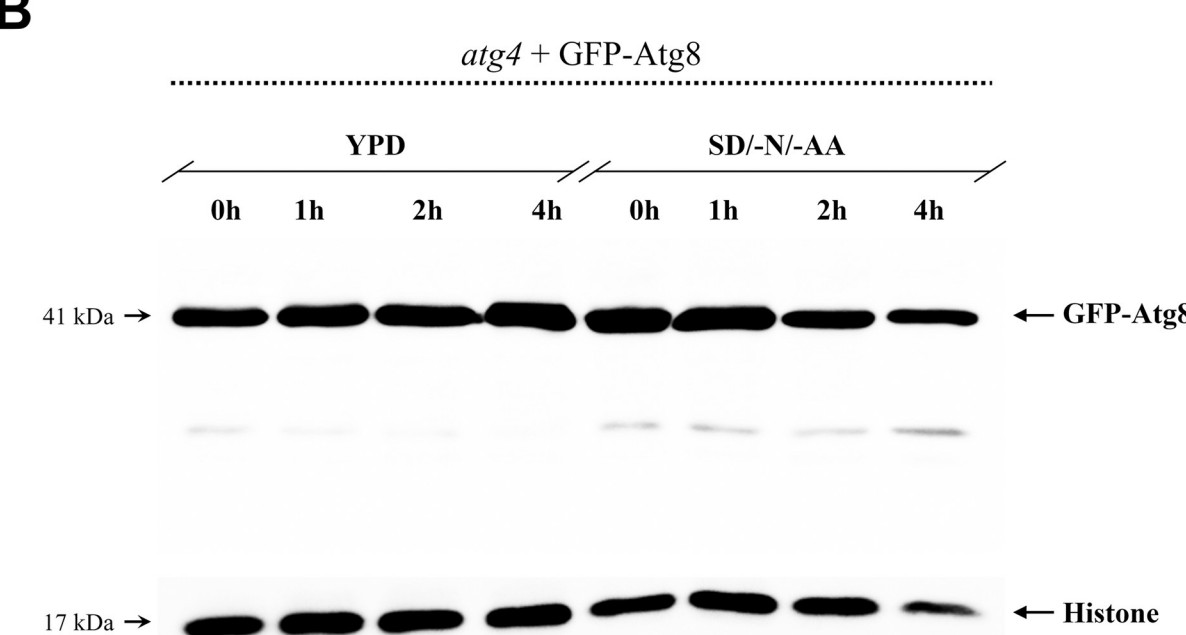

**Fig 4. Atg8 vacuolar degradation during autophagy in *C. neoformans*.** (**A**) Wild type (KN99α) and (**B**) *atg4* expressing GFP-Atg8 were grown in a YPD and SD/-N/-AA (nitrogen starvation) broth from 0–4 h. Western blot analysis incubated with primary antibody anti-GFP (1:2,000); n = 3 independent experiments. 41 kDa: full-length Atg8 fused with GFP. 27 kDa: free GFP generated by Atg8 vacuolar proteolysis. Histone was used as an internal reference (anti-histone, 1:2,000).

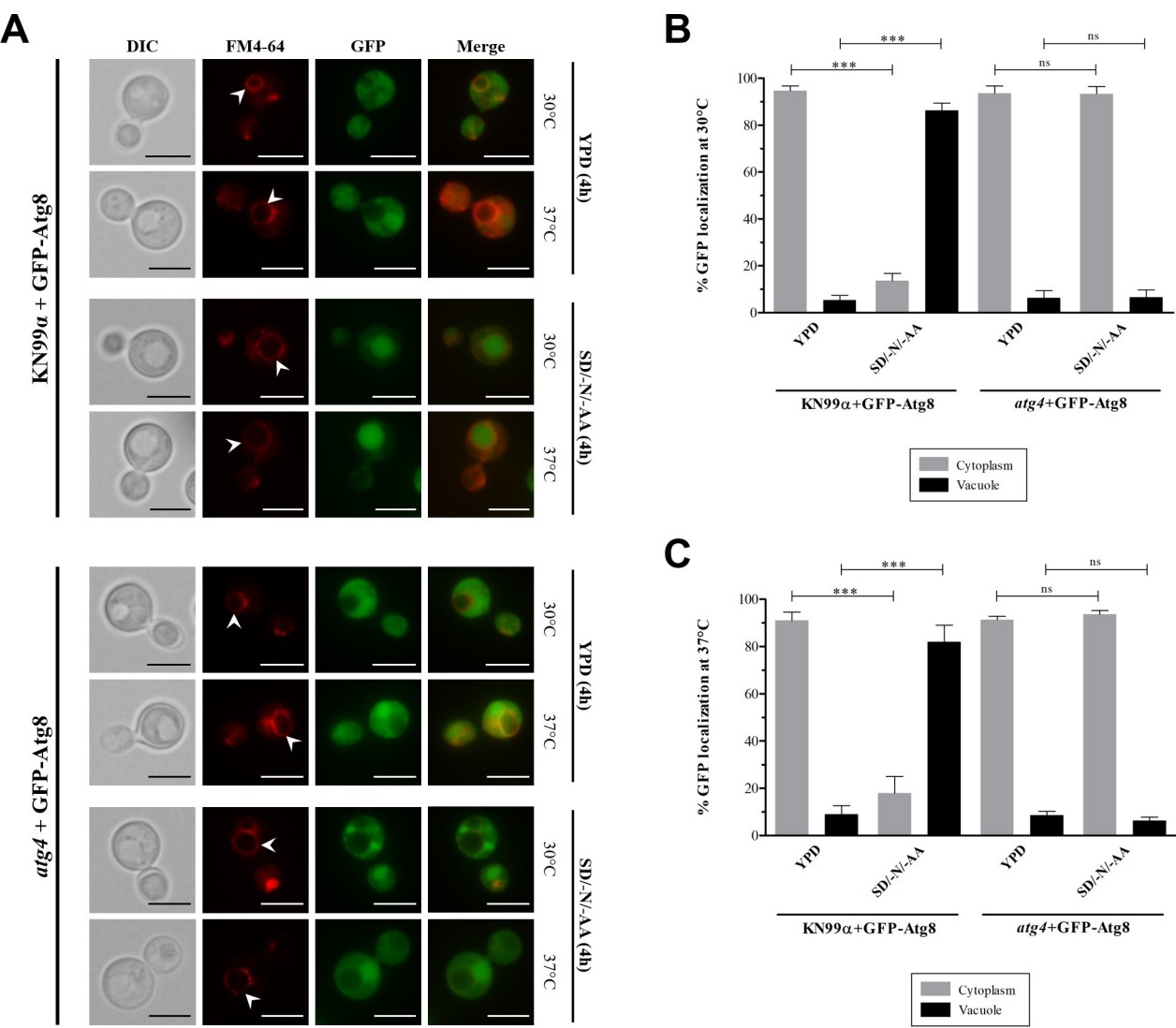

**Fig 5. GFP-Atg8 localization by fluorescence microscopy.** KN99α+GFP-Atg8 and *atg4*+GFP-Atg8 strains were cultured in YPD and SD/-N/-AA at 30˚C and 37˚C for 4 h. Vacuole labeled with FM4-64 (Invitrogen) dye. (**A**) GFP-Atg8 localization during autophagy pathway. White arrows indicate the vacuole membrane. Scale bars represent 5 μm. (**B**) Percentage of GFP-Atg8 localization at 30˚C. (**C**) Percentage of GFP-Atg8 localization at 37˚C. Images were captured with 100× objective lens. n = 3 independent experiments. Error bars represent the standard deviation. Statistical analysis performed using two-way ANOVA: *** ($P < 0.001$); ns (not significant).

revealed that most of the GFP-Atg8 was distributed in the cell cytoplasm. However, after autophagy induction by starvation, we observed a significant translocation of GFP-Atg8 to the vacuolar lumen at 30˚C and 37˚C (Fig 5A). In contrast, for the *atg4* mutants cultured in YPD, Atg8 fused with GFP was dispersed in the cytoplasm. On the other hand, when these cells lacking the protein Atg4 were cultured in SD/-N/-AA, we observed that GFP-Atg8 remains localized in the cytoplasm, demonstrating that Atg4 is required for the shifting of Atg8 to vacuole lumen (Fig 5A). When comparing the GFP localization according to the temperature, we observed a similar percentage of GFP-Atg8 in cells growing at 30˚C and 37˚C; in which for the wild type 86.3% and 82.3% of GFP-Atg8 were located in the cytoplasm (30˚C and 37˚C, respectively) and for *atg4* mutant 93.3% and 93.7% were located intravacuolar (30˚C and

37˚C, respectively). The data were statistically significant only for the wild type strains, for which we observed for cells cultured in YPD an average of 93% of GFP localized in the cytoplasm and, for cells growing in SD/-N/-AA, we found around 84% of GFP-Atg8 localized intravacuolar (Fig 5B and 5C).

Together, these results support the idea that Atg4 and Atg8 proteins in *C. neoformans* are involved in an autophagic mechanism, since the lack of both proteins impairs yeast growth in an autophagy induced environment. Moreover, the *ATG4* gene deletion affects the vacuolar breakdown of Atg8, indicating that Atg8 was not processed and could not form the autophagosome vesicle during the autophagy pathway. Additionally, the correct Atg8 intravacuolar localization occurs in an Atg4-dependent manner.

## Requirement of autophagy-related genes during stress growth condition

Recently it was shown that in *C. neoformans* functional autophagy pathway is important to several biological events related to pathogenesis [30–32]. To better characterize the effect of a defective autophagy process, we performed experiments to evaluate the growth phenotype in multi-stress conditions for the wild type (KN99α), *atg4* (two independent strains), *atg8* and reconstituted *atg8+ATG8*. The stress conditions tested were those performed by many authors to access the effect of several mutations throughout the *C. neoformans* genome that are thought to be important for pathogenesis [50,51]. For this analysis, we could not observe any phenotypic difference for the mutants growing under starvation (SD+/-N+/-AA), alkaline (pH), osmotic (KCl), saline (NaCl), nitrosative (NaNO$_2$), and cell membrane integrity (SDS) stress (data shown in S7 Fig). However, compared to the wild type, it was possible to notice growth differences between *atg4* and *atg8* strains in response to YPD supplemented with H$_2$O$_2$ and Congo Red, as shown in Fig 6. The *ATG4* and *ATG8* gene deletion resulted in an increased tolerance to hydrogen peroxide, supposedly affecting the fungal antioxidant defense and improving the yeast growth in the presence of H$_2$O$_2$. For the oxidative stress experiment, the *atg4* and *atg8* showed a growth improvement on YPD with 4 mM H$_2$O$_2$ at 30˚C. When incubating the cells at 37˚C, we observed for the autophagy mutants an increment of resistance to hydrogen peroxide on YPD containing 3 mM H$_2$O$_2$ and a slight cell growth on 4 mM H$_2$O$_2$, when compared to wild type strain (Fig 6A). Furthermore, the *atg4* and *atg8* growth was impaired by incubating the cells on YPD supplemented with 0.65% and 0.75% Congo Red at 30˚C and 37˚C. Therefore, the disruption of the autophagy pathway somehow impairs *C. neoformans* to efficiently respond to this stressful condition (Fig 6B).

It is known that the autophagy and ubiquitin-proteasome pathways (UPP) help to maintain the amino acid homeostasis in the cell [52–54]. Ding et al. [32] described that disrupting the UPP by the proteasome inhibitor bortezomib (BTZ at 50 μg/mL) had an amplified effect upon autophagy mutants when growing under nitrogen starvation. Therefore, we decided to evaluate the impact of BTZ presence when growing *atg4* and *atg8* mutants under nitrogen depletion medium. As described previously by Ding et al. [32] and observed in our analysis, BTZ reduced the growth pattern of the wild type strain (KN99α) after 48 hours under nitrogen deprivation, but it was not a significant decrease in the CFU (Fig 6C). However, the growth impairment was rather pronounced for the *atg8* mutant treated with BTZ, which had the growth reduced to 88% compared with the same strain without treatment. As expected, this impact was remediated when a wild type copy of *ATG8* was reintroduced in the *atg8* mutant, confirming the results published by Ding et al. [32]. Fig 6C shows that the two *atg4* independent mutants have significantly reduced their proliferation by 88% and 93% in the presence of BTZ, compared with the same strain without BTZ. This experiment allows us to suggest that the lack of Atg4 and Atg8 impairs the progression of the autophagy mechanism; additionally,

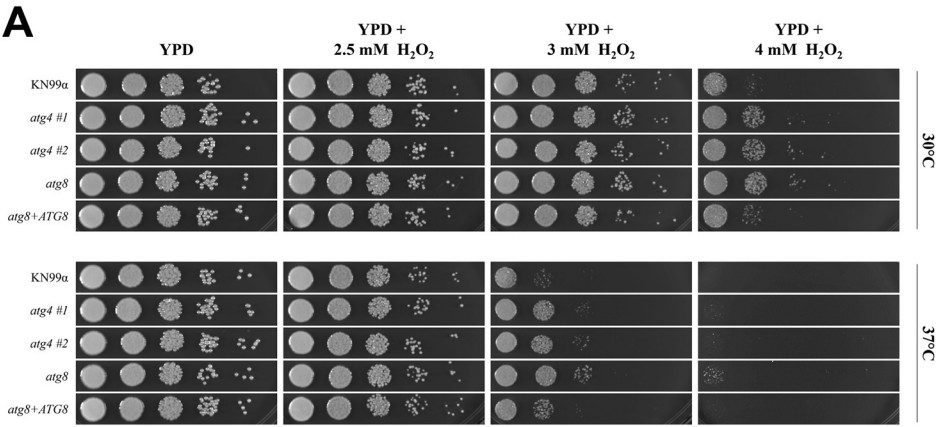

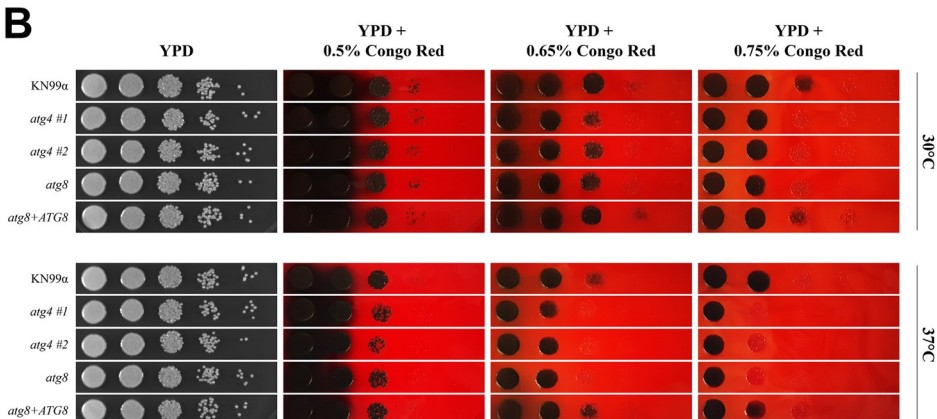

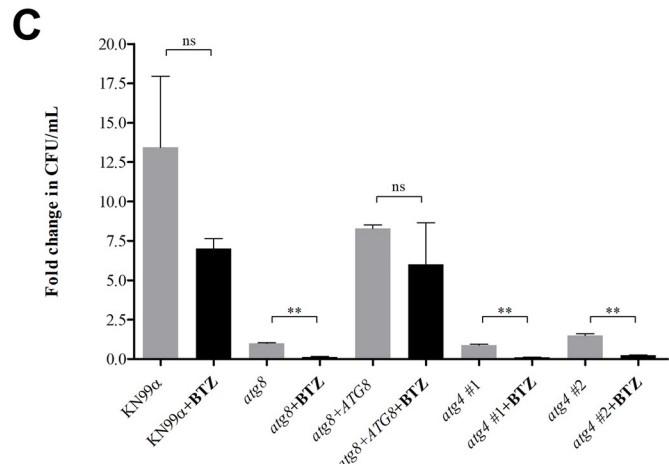

**Fig 6. Biological impact of *ATG4* (CNAG_02662) and *ATG8* (CNAG_00816) gene deletion on *C. neoformans*.**
Analysis of non-autophagic functions for KN99α (wild type); *atg4* (two independent mutants); *atg8; atg8+ATG8*. (**A**)
Oxidative stress phenotype. Growth analysis on YPD supplemented with 2.5–4 mM $H_2O_2$. Yeast cells were spotted by
10-fold serial dilutions. Plates were incubated at 30˚C and 37˚C for 48 h. (**B**) Cell wall integrity phenotype. Growth
analysis on YPD supplemented with 0.5%-0.75% Congo Red. Yeast cells were spotted by 10-fold serial dilutions. Plates
were incubated at 30˚C and 37˚C for 48 h. (**C**) Bortezomib (BTZ) sensitivity upon autophagy mutants. Fold change in
CFU/mL after 48 hours of induction in nitrogen-starvation medium (SD/-N/-AA), with or without BTZ (50 μg/mL).
Values shown are means of the experiments. Error bars represent the standard deviation. Unpaired *t* test: ** ($P < 0.05$);
ns (not significant). All the experiments are n = 2 independent replicates.

the deficiency of autophagy in combination with the inhibition of proteasome formation causes an unbalance in the amino acid homeostasis, leading to a drastic drop on nutrients recycling, that might impairs the cell to grow and/or survive under nitrogen deprivation.

Overall, these results indicate that in *C. neoformans* the autophagy mechanism may prepare the cell to respond to adverse environmental conditions by recycling nutrients likely, contributing to yeast survival.

## Discussion

Due to the fact that *C. neoformans* causes infections mainly in immunodeficient individuals [55], affecting up to 223,000 individuals per year [56] and that there is a constraint regarding treatment; it is necessary to broad the knowledge on this fungus biology in order to search for alternatives for treatment and disease control. Understanding the molecular mechanisms that connect both pathogenicity and autophagy could provide us new insights into the relationship established during the fungal infection [57]. One of the central steps of the autophagy pathway is the autophagosome formation in which, in yeast model *S. cerevisiae*, Atg4 is important to start the autophagosome synthesis by processing Atg8 [58]. Therefore, we characterized in the present study the autophagy-related proteins Atg4 and Atg8 in *C. neoformans*, hoping to clarify new information about their roles in the biology of this pathogenic fungus.

The multiple protein sequence alignment revealed that Atg8 in *C. neoformans* contains the C-terminal glycine (Gly117) that is highly conserved in the fungi species analyzed in our study, and also reported in other organisms like *Bipolaris maydis*, *Schizosaccharomyces pombe*, *Arabidopsis thaliana*, *Rattus norvegicus*, *Homo sapiens*, etc [59,60]. Moreover, we found by two-hybrid analysis that *C. neoformans* Atg8 interacts with Atg3, Atg4 and Atg7. In *S. cerevisiae*, Atg8 is processed in the C-terminal region by Atg4, to expose the glycine residue; then processed Atg8 is activated by an E1-like enzyme Atg7 and, subsequently, transferred to the E2-like enzyme Atg3, to promote the Atg8 conjugation with PE (Atg8-PE). Atg8-PE is essential for the double membrane autophagosome formation [58]. Among the three confirmed interactions, we decided to better investigate the autophagy-related protein 4.

Atg4 is a cysteine protease of Clan CA and family C54 that is a crucial for both Atg8-PE conjugation and subsequent delipidation [61]. Atg4 was first identified in yeast by Lang et al. [46], who demonstrated that this protein was essential for the process of delivering the autophagic vesicles to the vacuole lumen in *S. cerevisiae*. Since then, Atg4 has been an important target to study autophagy in different eukaryotic organisms. In our analysis of Atg4 we observed that, as in *S. cerevisiae*, the *C. neoformans* cysteine protease has the conserved domain of Peptidase_C54 Superfamily. The protease database *MEROPS* [62] describes the family C54 as a group of endopeptidases that hydrolases a peptide bond, with specificity to glycyl bonds. Furthermore, in *S. cerevisiae*, the catalytic residue of cysteine (Cys147) was identified as responsible for initially processing Atg8 at the C-terminus, exposing the glycine residue and allowing Atg8 to conjugate with PE (Atg8-PE) [63]. Comparing the Atg4 multiple sequence alignment with fungal species, the *C. neoformans* putative catalytic residue of cysteine (Cys568) is conserved. Other authors studying alignments of Atg4 homologues found that the Cys residue was highly conserved among different species, beyond those presented in this study, including *Botrytis cinerea*, *Candida glabrata*, *Fusarium graminearum*, *Penicillium oxalicum*, *Arabidopsis thaliana*, *Mus musculus*, *Homo sapiens*, and others [20,24,26].

Evidence that Atg4 and Atg8 in *C. neoformans* are highly conserved led us to validate their functions by heterologous complementation in *Saccharomyces*. In fungi, functional complementation of autophagy genes in *Saccharomyces* mutants was demonstrated for *Botrytis cinerea* [24,25], *Magnaporthe oryzae* [26], and *Sordaria macrospora* [23]. Here, we analyzed

the growth viability of the mutants and the complemented strains after cultivating for 20 days in starvation medium (SD/-N/-AA). In general, *S. cerevisiae atg* null mutants tend to lose their ability to survive in nitrogen starvation conditions [48]. In our study we observed that the *S. cerevisiae atg4Δ* and *atg8Δ* strains had defective growth in SD/-N/-AA. However, the phenotypes were recovered to the wild type level, once each mutant was complemented with their respective *C. neoformans* homologous gene. The same was observed when evaluating specifically the *C. neoformans* autophagy mutated strains, in that *atg4* and *atg8* also show a sensitivity phenotype after cultivated in depleted nutrient condition. In addition, Gontijo et al. [31] studying 21 putative genes involved in autophagy, showed that in *C. neoformans* the *ATG4* and *ATG8* gene transcription is induced by nitrogen starvation at 30˚C and 37˚C. Onodera and Ohsumi [52] reported that the loss of cellular viability in these *atg* mutants, cultured in nutritional deprivation, may be due to failure to sustain the amino acids supply, leading to inability to synthesize new proteins. Besides that, Suzuki et al. [64] presented that the cell death of *atg* mutants is in consequence of deficiency in respiratory function. The authors showed that, under starvation condition, the autophagy-defective mutants tend to accumulate reactive oxygen species (ROS), causing inhibition of mitochondrial DNA replication and resulting in cells unable to maintain mitochondrial function.

In most eukaryotic organisms, Atg8 is a central protein used to infer the progress of autophagy. Based on that, a reliable method to better understand the autophagy molecular mechanism is through monitoring the vacuolar delivery and proteolysis of GFP-Atg8 [65]. Here, we demonstrated that the fusion protein GFP-Atg8 can be also used to monitor the autophagy pathway in *C. neoformans*. Our western blot analysis revealed that Atg4 is essential for Atg8 vacuolar proteolysis, as the observations evidenced in yeast model *S. cerevisiae* [66]. Confirming our findings, the results obtained previously by other authors show that although Atg8 is expressed in the *atg4* mutant cells, the protein cannot be processed in the C-terminal region to form the Atg8-PE conjugate, interrupting the autophagosomal membrane synthesis and, by that, blocking the GFP-Atg8 degradation in the vacuole lumen [66,67]. Furthermore, fluorescence microscopy analysis allowed us to observe the GFP-Atg8 flux during nitrogen starvation. We demonstrated in *C. neoformans* that GFP-Atg8 is localized inside the vacuole in an Atg4-dependent manner. As previously published for *S. cerevisiae*, in wild type cells growing in a normal autophagy condition, GFP-Atg8 can be visualized as a puncta close to the vacuole, indicating that Atg8 is located in the pre-autophagosomal structure (PAS), in which the Atg-related proteins are recruited for autophagosome formation [68]. Although we could not see in *C. neoformans* a perivacuolar puncta signal for GFP-Atg8 when KN99α cells were cultured in starvation medium (SD/-N/-AA), we believe that the starvation induction for a period of 4 hours provided us with information on the later steps of the autophagy pathway. In this case, the GFP signal was concentrated within the vacuole, suggesting that the autophagosome double membrane was formed, fused with the vacuole, and lastly, the inner autophagosome membrane covered with GFP-Atg8-PE was delivered to the vacuole and degraded by vacuolar proteases [65].

Recently, new studies have reported that Atg proteins are involved in other biological process beyond autophagy. Oliveira et al. [30] showed that *ATG7* deletion in *C. neoformans* impairs not only the autophagy mechanism, but also affects the cellular size, reduces the pathogenicity in infected larvae, impairs the colonization in mice lung, and decreases the cell resistance to fungicidal activity of neutrophils. Ding et al. [32] found that *ATG1*, *ATG7*, *ATG8*, and *ATG9* genes in *C. neoformans* are involved in the autophagy mechanism and contribute to the virulence of the fungus. The authors observed that these mutants presented impaired maintenance of amino acids homeostasis under nitrogen starvation, reduced levels of virulence in a murine inhalation model, and contribute in different ways to the immune response and ability

to promote the virulence. In our study, we found that both *ATG4* and *ATG8* deletion in *C. neoformans* resulted in cells that were more resistant to oxidative stress when cultivated with hydrogen peroxide, defective growth in the presence of Congo Red maybe as a result of a cell wall integrity damage, and sensitivity to the proteasome inhibitor BTZ.

Several studies have found evidence for an interaction between autophagy and ROS. Most of the data in the literature have shown that autophagy can be also controlled by ROS, which are responsible for the modulation of survival or cell death cascades, under oxidative inductions [69]. Experiments performed in *Saccharomyces* by Pérez-Pérez et al. [20] show that Atg4 is one of the autophagy-related proteins controlled by reactive oxygen species. The authors observed that Atg4 activity is inhibited by oxidation, in a $H_2O_2$ concentration-dependent manner, thereby affecting the autophagic machinery. In most cases, basal levels of ROS play an important homeostatic role for cell survival but, at high levels, ROS are lethal to the cells [69]. For *C. neoformans* we observed that *atg4* and *atg8* mutants are tolerant to higher concentrations of hydrogen peroxide (3 and 4 mM $H_2O_2$) compared to the wild type strain. It is important to note that a similar finding was obtained in mouse embryo fibroblasts (MEFs) cells: *atg5* mutant cells were more resistant to cell death after exposure to $H_2O_2$, attributed by subsequent compensatory activation of extracellular signal-regulated kinases 1 and 2 (ERK1/2), which might interfere with cell survival against the oxidative damage [70].

Moreover, in *C. neoformans*, *atg4* and *atg8* mutants have an impaired growth in the presence of 0.65% and 0.75% Congo Red at 30˚C and 37˚C. Congo Red is cell wall-perturbing agent with high affinity for chitin, that causes weakness in the cell wall, activating the stress response [71]. For the endophytic fungus *Harpophora oryzae*, the *ATG5* gene displays an important role in autophagy, sporulation and cell wall integrity. In the presence of Congo Red the growth is highly inhibited in the *atg5* mutants, affecting the mycelial development [72]. Another finding in our investigation was the fact that *ATG4* and *ATG8* gene disruption resulted in sensitivity to the proteasome inhibitor BTZ. In accordance with our results, Ding et al. [32], studying the role of BTZ in the absence of a nitrogen source, reported that deletion of *ATG1*, *ATG7*, *ATG8* and *ATG9* genes in *C. neoformans* causes BTZ hypersensitivity cells, by inhibiting the ubiquitin-proteasome pathway. In addition, the authors pointed out that the autophagy mechanism is upregulated to compensate the proteasome inhibition; however, the effect of BTZ in *ATG* mutant cells is exacerbated when autophagy is compromised, resulting in a decrease in the availability of amino acids and other nutrients to the cell.

Taking these results together and based on *Saccharomyces* autophagy studies, we hypothesize the scenario that during the autophagy pathway in *C. neoformans*, Atg4 processes Atg8 by exposing the C-terminal glycine (Gly117). Thereby, processed Atg8 interacts with Atg7 and Atg3 to promote its conjugation with PE (Atg8-PE), allowing the double membrane autophagosome assembly/expansion, followed by the autophagosome-vacuole fusion and delivery of cargo to the vacuolar lumen. However, further experiments need to be developed to confirm our hypothesis. In conclusion, our study suggests that for *C. neoformans* the proteins Atg4 and Atg8 are involved in an autophagy mechanism under starvation induced conditions. The results also show that functional autophagy pathway may be part of distinct biological process in *C. neoformans*.

## Supporting information

**S1 Raw images.**
(PDF)

**S1 Table. Yeast strains used in this work.**
(PDF)

**S2 Table. List of oligonucleotides used in this study.**
(PDF)

**S1 Fig. Evolutionary analysis of the amino acid sequence with members of Atg8 family in different species of fungi.** The tree was constructed by the Neighbor-Joining method using the software MEGA v. 10.0.5. The evolutionary distances were computed using the Poisson correction method. The values of the branches indicate the percentages of replicates in which the associated taxa clustered in the bootstrap analysis (1,000 replicates).
(TIF)

**S2 Fig. Multiple alignment of homologous amino acid sequences of the Atg8 family.** Alignment using the ClustalW program [33]. Identical amino acids in all sequences are indicated in gray. Conserved glycine (G) residues in the C-terminal region are highlighted with a black background. The Ubiquitin-like superfamily domain is indicated by the dashed line. Filled arrows = predicted Atg4 protein binding sites; unfilled arrows = predicted Atg7 protein binding sites. Species and protein accession number: Sc = *Saccharomyces cerevisiae* (YBL078C); Ca = *Candida albicans* (CAWG_00835); Cn = *Cryptococcus neoformans* (CNAG_00816); Cg = *Cryptococcus gattii* (CGB_A9330C); Nc = *Neurospora crassa* (NCU01545); Af = *Aspergillus flavus* (AFLA_022400); Mo = *Magnaporthe oryzae* (MGG_01062); Um = *Ustilago maydis* (UMAG_05567).
(TIF)

**S3 Fig. Bait and prey autoactivation analysis.** The cloned vectors do not autonomously activate the reporter genes. (**A**) Growth of Y2HGold yeast strain transformed with the indicated bait (pGBKT7-*ATG8*). (**B**) Growth of Y2HGold yeast strain transformed individually with the indicated preys (pGADT7-*ATG3*/*ATG4*/*ATG7*). Two independent clones were tested in a total of n = 2 independent experiments. Plates incubated for 5 days at 30˚C. Assay performed with the Matchmaker™ Gold Yeast Two-Hybrid System (Clontech). SD: synthetic dextrose medium; Leu: leucine; Trp: tryptophan; X: 40 µg/mL X-α-Gal; A: 200 ng/mL Aureobasidin A; (+): pair of plasmids pGBKT7-53 and pGADT7-T used as positive control.
(TIF)

**S4 Fig. *In silico* analysis of Atg4 from *C. neoformans*.** (**A**) Atg4 from *S. cerevisiae* (YNL223W); 494 amino acids. (**B**) Atg4 from *C. neoformans* (CNAG_02662); 1,185 amino acids. Illustration of the protein domains using the IBS 1.0.3 software (CUCKOO Workgroup). Domains identified using the NCBI platform (Conserved Domain Database) [73]. (**C**) Multiple alignment of Atg4 homologous amino acid sequences. Region surrounding the active cysteine residue. Identical amino acids in all sequences are highlighted in gray. Catalytic residue of cysteine among the different members of the Atg4 family is indicated in black. Alignment using the ClustalW program [33]. Species and GenBank accession number: Sc = *Saccharomyces cerevisiae* (YNL223W); Cn = *Cryptococcus neoformans* (CNAG_02662); Cg = *Cryptococcus gattii* (CGB_K2500C); Nc = *Neurospora crassa* (NCU02433); Af = *Aspergillus flavus* (AFLA_104050); Mo = *Magnaporthe oryzae* (MGG_03580); Um = *Ustilago maydis* (UMAG_05142).
(TIF)

**S5 Fig. Deletion of the *ATG4* (YNL223W) and *ATG8* (YBL078C) genes in *Saccharomyces cerevisiae* (BY4741).** (**A**) Schematic representation of *ATG4* deletion by homologous recombination. (**B**) Confirmatory deletion performed by diagnostic PCR. Atg4ScF + Atg4ScR: *atg4Δ* amplicon with 2,927 bp; Atg4ScF + KanMXR: *atg4Δ* amplicon with 610 bp. Colonies 1, 2, 4

and 5 are *atg4Δ*. (**C**) Schematic representation of *ATG8* deletion by homologous recombination. (**D**) Confirmatory deletion performed by diagnostic PCR. Atg8ScF + Atg8ScR: *atg8Δ* amplicon with 2,889 bp; Atg8ScF + KanMXR: *atg8Δ* amplicon with 587 bp. Colonies 1, 2 and 5 are *atg8Δ*. Atg4ScF, Atg4ScR, Atg8ScF, Atg8ScR and KanMXR: primers used in PCR. Lanes 1–5: colonies growing in selective medium. KanMX6: G418 resistance. WT: wild type BY4741 gDNA, (–): negative control. 1 Kb+: GeneRuler 1 Kb Plus DNA Ladder (Thermo Fisher Scientific).
(TIF)

**S6 Fig. Deletion of the *ATG4* (CNAG_02662) and *ATG8* (CNAG_00816) genes in *Cryptococcus neoformans* (KN99α).** (**A**) Strategy to confirm *ATG4* deletion by double-joint methodology. (**B**) Southern blot analysis confirming *ATG4* deletion. Lanes 1–6: colonies growing in selective plates. Genomic DNA digested using *Eco*RI (G^AATTC) restriction enzyme. Restriction fragment sizes expected after the hybridization signal: KN99α with 5,943 bp and *atg4* with 1,032 bp, as indicated in the schematic representation. Transformants 1–6 are *atg4*. (**C**) Strategy to confirm *ATG8* deletion by double-joint methodology. (**D**) Southern blot analysis confirming *ATG8* deletion. Lanes 1–4: colonies growing in selective plates. Genomic DNA digested using *Eco*RV (GAT^ATC) restriction enzyme. Restriction fragment sizes expected after the hybridization signal: KN99α with 1,996 bp and *atg8* with 6,150 bp, as indicated in the schematic representation. Transformants 2 and 4 are *atg8*. Hyg$^R$: hygromycin resistance. KN99α: wild type gDNA. Arrows indicate the molecular weights based on GeneRuler 1 Kb DNA Ladder (Fermentas).
(TIF)

**S7 Fig. Evaluation of wild type (KN99α) and *atg4* and *atg8* growth phenotype under multistress conditions.** (**A**) Starvation stress (SD+/-N+/-AA). (**B**) Alkaline stress (pH). (**C**) Osmotic and saline stress (KCl and NaCl, respectively). (**D**) Nitrosative and cell membrane integrity stress (NaNO$_2$ and SDS, respectively). Yeast cells were spotted by 10-fold serial dilutions. The plates were incubated at 30˚C and 37˚C for 48 h. All the experiments are n = 2 independent replicates.
(TIF)

## Acknowledgments

The authors would like to thank Accord/Intas Pharmaceuticals for kindly providing the proteasome inhibitor bortezomib.

## Author Contributions

**Conceptualization:** Renata Castiglioni Pascon, Alexander Idnurm, Marcelo Afonso Vallim.

**Data curation:** Thiago Nunes Roberto, Ricardo Ferreira Lima.

**Formal analysis:** Thiago Nunes Roberto, Ricardo Ferreira Lima, Marcelo Afonso Vallim.

**Funding acquisition:** Renata Castiglioni Pascon, Alexander Idnurm, Marcelo Afonso Vallim.

**Investigation:** Thiago Nunes Roberto, Ricardo Ferreira Lima, Renata Castiglioni Pascon, Alexander Idnurm, Marcelo Afonso Vallim.

**Methodology:** Thiago Nunes Roberto, Ricardo Ferreira Lima, Renata Castiglioni Pascon, Alexander Idnurm, Marcelo Afonso Vallim.

**Project administration:** Marcelo Afonso Vallim.

**Resources:** Renata Castiglioni Pascon, Alexander Idnurm, Marcelo Afonso Vallim.

**Software:** Thiago Nunes Roberto, Ricardo Ferreira Lima.

**Supervision:** Alexander Idnurm, Marcelo Afonso Vallim.

**Validation:** Thiago Nunes Roberto, Marcelo Afonso Vallim.

**Visualization:** Thiago Nunes Roberto, Marcelo Afonso Vallim.

**Writing – original draft:** Thiago Nunes Roberto, Marcelo Afonso Vallim.

**Writing – review & editing:** Ricardo Ferreira Lima, Renata Castiglioni Pascon, Alexander Idnurm.

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
