## [Decision Letter · Decision Letter 0]

13 Jan 2020

PONE-D-19-32465

Biological functions of the Atg4 and Atg8 autophagy pathway components in Cryptococcus neoformans

PLOS ONE

Dear Dr Vallim,

Thank you for submitting your manuscript to PLOS ONE. After careful consideration, we feel that it has merit but does not fully meet PLOS ONE’s publication criteria as it currently stands. Therefore, we invite you to submit a revised version of the manuscript that addresses the points raised during the review process.

Reviewer 1:

This manuscript reported the roles of Atg4 and Atg8 in Cryptococcus neoformans. The authors found that Atg4 interacts with Atg8 and controls Atg8 localization and degradation. In addition, the author presented that Atg4 and Atg8 are required for nitrogen starvation and stress response. This manuscript contains meaningful results to understand for autophagy process and fungal biology. However, this manuscript should be improved before publication in PLoS One. 

- The Atg8-related results in Figures 5 and 8 have already been published elsewhere, so it may be worth focusing on Atg4 in this paper. 

- Lines 324-348, it is questionable whether this section is required for this manuscript. Atg8 has been characterized other study. In Figure 1, the authors just aligned several sequences, but detailed results about the function of this domain or residues have not been studied in this manuscript. Therefore, Figure 1 may not be necessary for in this manuscript.

- Lines 361-362, In this section, the authors just checked protein interaction, but not autophagosome synthesis. So please revise this subtitle. 

- Lines 390-392, Please provide proper reference or result.

- Figure 3, it may not be necessary in the main body and can be moved to Supplementary figure.

- Lines 447-462, While the authors mentioned that they used two independent mutants, Atg4 complemented strains should be needed.

- Lines 477-481, the authors just checked Atg8 degradation, but we cannot know Atg8 degradation in vacuolar lumen.

- Localization of Atg4 can be examined.

Reviewer 2:

This is an interesting and well-written manuscript.  Whilst the findings are not, perhaps, that surprising, it nonetheless reports a helpful and important advance for the field.  The work builds on previous findings from Ding and colleagues (reference 32) and characterises Atg4 and Atg8 in the human pathogenic fungus Cryptococcus neoformans.

Overall, I think this is an interesting, high-quality paper and have only relatively minor comments.

1. The title is somewhat misleading.  Whilst the authors characterise Atg4/8, they do not really address the question of autophagy within the manuscript.  Consequently, I would recommend rephrasing to “Biological functions of Atg4 and Atg8 in Cryptococcus neoformans”

2. Line 72: does blocking autophagy actually seem like a viable treatment for invasive fungal disease? This is also mentioned in the discussion. Surely any attempt to block fungal autophagy would also impact on host autophagy.  I would recommend rephrasing.

3. Figure 2. The yeast 2 hybrid experiments should include additional controls to ensure ATG 8 is specifically binding to ATG3,4, 7 and not indiscriminately to other proteins.  For instance, have other ATG proteins been tested?   

4. Figure 6.  Was this a single experiment or a representative of multiple replicates? If so, how many?

5. Line 456: “For the atg4 mutants we tested two independent strains, since we did not manage to restitute the wild type copy back into the mutant.”  This is intriguing – can the authors speculate why complementation was so challenging?  Is there evidence that atg4 overexpression may be lethal, for instance?  

6. Line 607: remove “basically”

7. Line 682: “puncta” should be “punctate”, I think?

8. Line 690: change “others” to “other”

9. Line 743: development and pathogenesis were not examined here, so this comment should be removed.

10. Ensure that the number of technical and biological replicates is clearly stated in all figure legends.  In some cases the information is there (e.g. “biological triplicate”) but would be better expressed as “n=3 independent experiments”.

Reviewer 3:

In this work entitled “Biological functions of the Atg4 and Atg8 autophagy pathway components in Cryptococcus neoformans” Nunes Roberto and co-workers described the that Atg4 and Atg8 are conserved proteins that interact physically with each other. ATG gene deletions resulted in cells sensitive to nitrogen starvation. ATG4 gene disruption affects Atg8 degradation and its translocation to the vacuole lumen, after autophagy induction. The mutants showed a more resistant phenotype to oxidative stress compared to the wild type and had an impaired growth in the presence of the cell wall perturbing agent Congo Red, and are sensitive to the proteasome inhibitor bortezomib (BTZ). The authors conclude based on their results that the autophagy-related proteins Atg4 and Atg8 play an important role in the autophagy pathway; which are required for autophagy regulation, maintenance of amino acid levels and cell adaptation to stressful conditions in C. neoformans.

The manuscript is well written, and the experiments interpretation support the authors hypothesis. 

Minor comments:

Line 37 - Delete OF from the sentence “responsible for delivering OF the cargo from the cytoplasm to the vacuole lumen.”

Line 502 – “But” is an informal adverb, the authors should use a more formal one.

Lines 594 -596 – This sentence is confusing, consider revising.

Discussion Lines 599 – 612  - This first paragraph is a little repetitive, because iit is well described in the introduction.

Line 661 – there is verb conjugation problem – the authors should use SHOWED or HAD SHOWN.

The authors should perform infection assays with the mutants to show if there is any alteration.

Line 684 – “we believe that the starvation induction for a period of 4 hours provided us information on the later steps of the autophagy pathway”. Did the authors perform different time point analysis?

Figure 5 - The authors should discuss in more details the smaller number of cells in the mutants, they should perform a growth curve of the mutants growing in distinct conditions.

In figure 8 no visual difference in growth is observed, that´s why a growth curve would be useful.  

Figure 6 - determine the intensity of the bands in order to have a quantitative analysis of the degradation.

We would appreciate receiving your revised manuscript by Feb 27 2020 11:59PM. To enhance the reproducibility of your results, we recommend that if applicable you deposit your laboratory protocols in protocols.io, where a protocol can be assigned its own identifier (DOI) such that it can be cited independently in the future. For instructions see: http://journals.plos.org/plosone/s/submission-guidelines#loc-laboratory-protocols

We look forward to receiving your revised manuscript.

Kind regards,

Vladimir Trajkovic

Academic Editor

PLOS ONE

Journal Requirements:

Additional Editor Comments (if provided):

Reviewers' comments:

Reviewer's Responses to Questions

**Comments to the Author**

1. Is the manuscript technically sound, and do the data support the conclusions?

Reviewer #1: Partly

Reviewer #2: Yes

Reviewer #3: Yes

2. Has the statistical analysis been performed appropriately and rigorously? 

Reviewer #1: Yes

Reviewer #2: Yes

Reviewer #3: Yes

3. Have the authors made all data underlying the findings in their manuscript fully available?

Reviewer #1: Yes

Reviewer #2: Yes

Reviewer #3: Yes

4. Is the manuscript presented in an intelligible fashion and written in standard English?

Reviewer #1: No

Reviewer #2: Yes

Reviewer #3: Yes

5. Review Comments to the Author

Reviewer #1: This manuscript reported the roles of Atg4 and Atg8 in Cryptococcus neoformans. The authors found that Atg4 interacts with Atg8 and controls Atg8 localization and degradation. In addition, the author presented that Atg4 and Atg8 are required for nitrogen starvation and stress response. This manuscript contains meaningful results to understand for autophagy process and fungal biology. However, this manuscript should be improved before publication in PLoS One.

- The Atg8-related results in Figures 5 and 8 have already been published elsewhere, so it may be worth focusing on Atg4 in this paper.

- Lines 324-348, it is questionable whether this section is required for this manuscript. Atg8 has been characterized other study. In Figure 1, the authors just aligned several sequences, but detailed results about the function of tis domain or residues have not been study in this manuscript. Therefore, Figure 1 may not be necessary for in this manuscript.

- Lines 361-362, In this section, the authors just checked protein interaction, but not autophagosome synthesis. So please revise this sub-title.

- Lines 390-392, Please provide proper reference or result.

- Figure 3, it may not be necessary in the main body and can be moved to Supplementary figure.

- Lines 447-462, While the authors mentioned that they used two independent mutants, Atg4 complemented strains should be needed.

- Lines 477-481, the authors just checked Atg8 degradation, but we can not know Atg8 degradation in vacuolar lumen.

- Localization of Atg4 can be examined.

Reviewer #2: Review of PONE-D-19-32465.

This is an interesting and well-written manuscript. Whilst the findings are not, perhaps, that surprising, it nonetheless reports a helpful and important advance for the field. The work builds on previous findings from Ding and colleagues (reference 32) and characterises Atg4 and Atg8 in the human pathogenic fungus Cryptococcus neoformans.

Overall, I think this is an interesting, high-quality paper and have only relatively minor comments.

1. The title is somewhat misleading. Whilst the authors characterise Atg4/8, they do not really address the question of autophagy within the manuscript. Consequently, I would recommend rephrasing to “Biological functions of Atg4 and Atg8 in Cryptococcus neoformans”

2. Line 72: does blocking autophagy actually seem like a viable treatment for invasive fungal disease? This is also mentioned in the discussion. Surely any attempt to block fungal autophagy would also impact on host autophagy. I would recommend rephrasing.

3. Figure 2. The yeast 2 hybrid experiments should include additional controls to ensure ATG 8 is specifically binding to ATG3,4, 7 and not indiscriminately to other proteins. For instance, have other ATG proteins been tested?

4. Figure 6. Was this a single experiment or a representative of multiple replicates? If so, how many?

5. Line 456: “For the atg4 mutants we tested two independent strains, since we did not manage to restitute the wild type copy back into the mutant.” This is intriguing – can the authors speculate why complementation was so challenging? Is there evidence that atg4 overexpression may be lethal, for instance?

6. Line 607: remove “basically”

7. Line682: “puncta” should be “punctate”, I think?

8. Line 690: change “others” to “other”

9. Line 743: development and pathogenesis were not examined here, so this comment should be removed.

10. Ensure that the number of technical and biological replicates is clearly stated in all figure legends. In some cases the information is there (e.g. “biological triplicate”) but would be better expressed as “n=3 independent experiments”.

Reviewer #3: In this work entitled “Biological functions of the Atg4 and Atg8 autophagy pathway components in Cryptococcus neoformans” Nunes Roberto and co-workers described the that Atg4 and Atg8 are conserved proteins that interact physically with each other. ATG gene deletions resulted in cells sensitive to nitrogen starvation. ATG4 gene disruption affects Atg8 degradation and its translocation to the vacuole lumen, after autophagy induction. The mutants showed a more resistant phenotype to oxidative stress compared to the wild type and had an impaired growth in the presence of the cell wall perturbing agent Congo Red, and are sensitive to the proteasome inhibitor bortezomib (BTZ). The authors conclude based on their results that the autophagy-related proteins Atg4 and Atg8 play an important role in the autophagy pathway; which are required for autophagy regulation, maintenance of amino acid levels and cell adaptation to stressful conditions in C. neoformans.

The manuscript is well written, and the experiments interpretation support the authors hypothesis.

Minor comments:

Line 37 - Delete OF from the sentence “responsible for delivering OF the cargo from the cytoplasm to the vacuole lumen.”

Line 502 – But is an informal adverb, the authors should use a more formal one.

Lines 594 -596 – This sentence is confusing, consider revising.

Discussion Lines 599 – 612 - This first paragraph is a little repetitive, because iit is well described in the introduction.

Line 661 – there is verb conjugation problem – the authors should use SHOWED or HAD SHOWN.

The authors should perform infection assays with the mutants to show if there is any alteration.

Line 684 – “we believe that the starvation induction for a period of 4 hours provided us information on the later steps of the autophagy pathway”. Did the authors perform different time point analysis?

Figure 5 - The authors should discuss in more details the smaller number of cells in the mutants, they should perform a growth curve of the mutants growing in distinct conditions.

In figure 8 no visual difference in growth is observed, that´s why a growth curve would be useful.

Figure 6 - determine the intensity of the bands in order to have a quantitative analysis of the degradation.

6. PLOS authors have the option to publish the peer review history of their article (what does this mean?). If published, this will include your full peer review and any attached files.

Reviewer #1: No

Reviewer #2: No

Reviewer #3: Yes: Lysangela Ronalte Alves

---

## [Author Response · Author response to Decision Letter 0]

19 Feb 2020

Reviewer #1

Reviewer comment:

(1) Comm.: “The Atg8-related results in Figures 5 and 8 have already been published elsewhere, so it may be worth focusing on Atg4 in this paper.”

Answer: Figure 5 (current Figure 3) shows that C. neoformans atg4 and atg8 mutants have their cellular viability decreased after 20 days of nitrogen starvation. Figure 8 (current Figure 6) shows C. neoformans atg mutants growth phenotype under stress condition. Figure 5 (current Figure 3) is an experiment conducted for C. neoformans strains, so far not published elsewhere. Figures 8A and B are not published elsewhere and Figure 8C (current Figure 6C) has been partially published by Ding et al. (2018), which have previously demonstrated that C. neoformans atg8 mutant is sensitive to Bortezomib (BTZ), we included the wild-type and atg8 mutant in our experiment and used the data in the graphic to validate our atg4 analysis. By saying that, we intend to keep both figures in the main body of this manuscript.

(2) Comm.: “Lines 324-348, it is questionable whether this section is required for this manuscript. Atg8 has been characterized other study. In Figure 1, the authors just aligned several sequences, but detailed results about the function of this domain or residues have not been studied in this manuscript. Therefore, Figure 1 may not be necessary for this manuscript.”

Answer: Indeed C. neoformans Atg8 has been studied previously by Hu et al. (2008) and Ding et al. (2018), but they did not study and correlate the domains with autophagy pathway. So far, the authors have identified and shown the role of Atg8 for virulence in a murine model of infection, importance to sustain the amino acid levels and atg8 mutant is sensitive to proteasome inhibitor Bortezomib (BTZ). Although our phylogenetic analysis showed that C. neoformans ATG8 did not cluster with Saccharomyces cerevisiae ATG8 (Supplementary Figure 1), we demonstrated in Figure 1 (current Supplementary Figure 2) that those proteins are highly conserved, presenting the predicted Atg4 and Atg7 proteins binding sites and C-terminal glycine residue, important in Saccharomyces during the autophagy pathway, allowing the conjugation of Atg8 with PE and, consequently, promoting the autophagosome synthesis. This information indicated us that C. neoformans Atg8 could be involved in autophagy mechanism and could work in a similar way to that reported for S. cerevisiae. We think that Atg8 characterization section is important and necessary for this study, so we choose to keep it in the main body of this manuscript. However, we moved the Figure 1 from the main body to Supplementary Figure section.

(3) Comm.: “Lines 361-362, in this section, the authors just checked protein interaction, but not autophagosome synthesis. So please revise this subtitle.”

Answer: We agree and the subtitle was edited and now one can read: “Interactions of Atg8 with autophagy-related proteins Atg3, Atg4 and Atg7”.

(4) Comm.: “Lines 390-392, please provide proper reference or result.”

Answer: We added the numbers of the figures for the results described in the sentence.

(5) Comm.: “Figure 3, it may not be necessary in the main body and can be moved to Supplementary figure.”

Answer: As suggested we moved the Figure 3 from the main body to Supplementary Figure section. 

(6) Comm.: “Lines 447-462, while the authors mentioned that they used two independent mutants, Atg4 complemented strains should be needed.”

Answer: We made several unsuccessful attempts to restitute back the wild-type native locus into the atg4 mutant and, therefore, we used two independent mutants in our analysis. This strategy has been used by other authors in the field, p.e. Fu et al. 2018. Genetics, 208: 639–653 (Heitman’s group), Martho et al. 2016. PLoS One, 11(10):e0163919 (Pascon’s group), among others.

(7) Comm.: “Lines 477-481, the authors just checked Atg8 degradation, but we cannot know Atg8 degradation in vacuolar lumen.”

Answer: As we did not perform an experiment to prove Atg8 degradation in vacuolar lumen, we revised the sentence and now one can read: “The KN99α data suggest a normal autophagic flux, with Atg8 being degraded by releasing the GFP fragment in the course of time. On the other hand, in the atg4 mutant only the full length GFP-Atg8 bands were detected (current Figure 4), suggesting that the autophagosome synthesis was blocked and, consequently, Atg8 was not degraded.”

(8) Comm.: “Localization of Atg4 can be examined.”

Answer: Atg4 is a cytoplasmic protein that is not carried along the autophagy pathway, unlike Atg8 protein that is transported from cytoplasm to vacuole. Therefore, by analyzing the literature, we believe that would not be enlightening to include in this study an experiment to verify the localization of the autophagy-related protein Atg4.

Reviewer #2 

Reviewer comment:

(1) Comm.: “The title is somewhat misleading. Whilst the authors characterize Atg4/8, they do not really address the question of autophagy within the manuscript. Consequently, I would recommend rephrasing to “Biological functions of Atg4 and Atg8 in Cryptococcus neoformans”.”

Answer: We changed the title as suggested, but we included the term “autophagy-related proteins”. Therefore, the title has been rephrased to “Biological functions of the autophagy-related proteins Atg4 and Atg8 in Cryptococcus neoformans”.

(2) Comm.: “Line 72: does blocking autophagy actually seem like a viable treatment for invasive fungal disease? This is also mentioned in the discussion. Surely any attempt to block fungal autophagy would also impact on host autophagy. I would recommend rephrasing.”

Answer: It is important to understand the autophagy mechanism in different species, since there are some autophagy-related genes uniquely codified in a single organism. It was previously reported by Gontijo et al. (2017) that some pathogenic fungi like C. neoformans, C. albicans and A. fumigatus have fewer autophagy genes not codified in their genome, when compared to S. cerevisiae. Besides that, in their review Lynch-Day and Klionsky (2010) discuss the idea that is important to identify receptors and unique factors occurring in the autophagy pathway in higher eukaryotes. So, when specific higher eukaryotes autophagy components are identified and differentiated from pathogenic microorganisms, blocking a particular fungal Atg protein could not impact on host autophagy and, consequently, could be a target for new antifungal treatment. The sentence was rephrased as suggest, to clarify the information above.

(3) Comm.: “Figure 2. The yeast 2 hybrid experiments should include additional controls to ensure ATG8 is specifically binding to ATG3,4, 7 and not indiscriminately to other proteins. For instance, have other ATG proteins been tested?”

Answer: In the current Supplementary Figure 3 we demonstrate that, individually, bait (pGBKT7-ATG8) and preys (pGADT7-ATG3/ATG4/ATG7) do not autonomously autoactivate the two-hybrid reporter genes, in the absence of an interactant protein. This result confirmed that Atg8 is not indiscriminately binding to other proteins and, by that, our yeast two-hybrid interactions results are validated.

(4) Comm.: “Figure 6. Was this a single experiment or a representative of multiple replicates? If so, how many?”

Answer: The Figure 6 (current Figure 4) is a representative of an experiment conducted three times independently. As suggested elsewhere, we included “n=3 independent experiments” in the figure legend.

(5) Comm.: “Line 456: “For the atg4 mutants we tested two independent strains, since we did not manage to restitute the wild type copy back into the mutant.” This is intriguing – can the authors speculate why complementation was so challenging? Is there evidence that atg4 overexpression may be lethal, for instance?”

Answer: We made several attempts without success to restitute back the wild-type native locus into the atg4 mutant and, therefore, we used two independent mutants in our analysis. There is no evidence in the literature on atg4 overexpression lethality. However, this far we do not have an explanation for our frustrated attempts, nonetheless we can speculate that the wild-type PCR fragment amplified could be too big (5,031 bp) for the homologous recombination, though experiments should be conducted to explore this speculation. Nevertheless, other groups have used independent mutants to validate their results without using a reconstituted strain, please see for example: Fu et al. 2018. Genetics, 208: 639–653 (Heitman’s group), Martho et al. 2016. PLoS One, 11(10):e0163919 (Pascon’s group).

(6) Comm.: “Line 607: remove “basically””

Answer: The word “basically” was removed from the sentence.

(7) Comm.: “Line 682: “puncta” should be “punctate”, I think?”

Answer: In the literature the perivacuolar accumulation of GFP-Atg8 is referred to as a fluorescent puncta signal. Therefore, we maintained the term “puncta”, but rephrased the words “puncta structure” to “perivacuolar puncta signal”.

(8) Comm.: “Line 690: change “others” to “other”.”

Answer: The word “others” was corrected to “other” in the sentence.

(9) Comm.: “Line 743: development and pathogenesis were not examined here, so this comment should be removed.”

Answer: We agree and removed the comment.

(10) Comm.: “Ensure that the number of technical and biological replicates is clearly stated in all figure legends. In some cases the information is there (e.g. “biological triplicate”) but would be better expressed as “n=3 independent experiments”.”

Answer: We have edited all the figures legends, expressing the numbers of replicates as suggested, to ensure that this information is clearly stated. 

Reviewer #3

Reviewer comment:

(1) Comm.: “Line 37 - Delete OF from the sentence “responsible for delivering OF the cargo from the cytoplasm to the vacuole lumen”.”

Answer: The preposition “of” was deleted from the sentence.

(2) Comm.: “Line 502 – “But” is an informal adverb, the authors should use a more formal one.”

Answer: We agree and changed the adverb to “However”.

(3) Comm.: “Lines 594-596 – This sentence is confusing, consider revising.”

Answer: We revised the sentence mentioned and now it reads: “Overall, these results indicate that in C. neoformans the autophagy mechanism may prepare the cell to respond to adverse environmental conditions by recycling nutrients likely, contributing to yeast survival”.

(4) Comm.: “Discussion Lines 599-612 – This first paragraph is a little repetitive, because it is well described in the introduction.”

Answer: The paragraph was rephrased.

(5) Comm.: “Line 661 – there is verb conjugation problem – the authors should use SHOWED or HAD SHOWN.”

Answer: We replaced the verb “shown” to “showed”.

(6) Comm.: “The authors should perform infection assays with the mutants to show if there is any alteration.”

Answer: We performed some inconclusive assays that showed for C. neoformans atg4 mutant no difference in virulence in Galleria mellonella. Besides that, Hu et al. (2008) and Ding et al. (2018) have already reported that C. neoformans atg8 mutant presented attenuated virulence in a BALB/c mouse host.

(7) Comm.: “Line 684 – “we believe that the starvation induction for a period of 4 hours provided us information on the later steps of the autophagy pathway”. Did the authors perform different time point analysis?”

Answer: When we first performed the experiment, it was tested the induction with 2 and 4 hours. But, with 2 hours we did not see any difference between the wild type and atg4 mutant fluorescence signals; the GFP-Atg8 signals was spread all over the cytoplasm for WT and mutant strains. So, when we increased the period of starvation induction to 4 hours, we saw the differences related in this manuscript (current Figure 5). Therefore, with this knowledge, we proceed the biological replicates considering only the inductions for a period of 4 hours.

(8) Comm.: “Figure 5 - The authors should discuss in more details the smaller number of cells in the mutants, they should perform a growth curve of the mutants growing in distinct conditions. In figure 8 no visual difference in growth is observed, that´s why a growth curve would be useful.”

Answer: The fourth paragraph in the discussion section presents more details about the smaller numbers of cells in C. neoformans atg4 and atg8 mutants under nitrogen starvation (Figure 5, current Figure 3). As observed in S. cerevisiae, “atg null mutants tend to lose their ability to survive in nitrogen starvation conditions”; C. neoformans “atg4 and atg8 also show a sensitivity phenotype after cultivated in depleted nutrient condition”. Based on information reported in the literature we discuss that the loss of cellular viability in atg mutants, growing in nutritional deprivation condition, could be due to failure to sustain the amino acids supply and/or to mitochondrial failure, causing deficiency in respiratory function. A visual difference in growth was observed in Figure 8 (current Figure 6) for atg4 and atg8 mutants growing in YPD with 3 mM H2O2 (37°C) and 4 mM H2O2 (30°C), and in YPD supplemented with 0.65% and 0.75% Congo Red (30°C and 37°C). The serial dilution spot test is better to visualize these results since hydrogen peroxide is an unstable compound and Congo Red is not used in growth curve.

(9) Comm.: “Figure 6 – determine the intensity of the bands in order to have a quantitative analysis of the degradation.”

Answer: The experiment performed was qualitative and aimed to monitor the GFP-Atg8 degradation. We do not intend to quantify the bands at this moment. In this case, the visual analysis of the free GFP bands was enough to demonstrate the Atg8 degradation under nitrogen starvation.

---

## [Decision Letter · Decision Letter 1]

13 Mar 2020

Biological functions of the autophagy-related proteins Atg4 and Atg8 in Cryptococcus neoformans

PONE-D-19-32465R1

Dear Dr. Vallim,

We are pleased to inform you that your manuscript has been judged scientifically suitable for publication and will be formally accepted for publication once it complies with all outstanding technical requirements.

With kind regards,

Vladimir Trajkovic

Academic Editor

PLOS ONE

Additional Editor Comments (optional):

Reviewers' comments:

Reviewer's Responses to Questions

**Comments to the Author**

1. If the authors have adequately addressed your comments raised in a previous round of review and you feel that this manuscript is now acceptable for publication, you may indicate that here to bypass the “Comments to the Author” section, enter your conflict of interest statement in the “Confidential to Editor” section, and submit your "Accept" recommendation.

Reviewer #1: All comments have been addressed

Reviewer #2: All comments have been addressed

Reviewer #3: All comments have been addressed

2. Is the manuscript technically sound, and do the data support the conclusions?

Reviewer #1: Yes

Reviewer #2: Yes

Reviewer #3: Yes

3. Has the statistical analysis been performed appropriately and rigorously? 

Reviewer #1: Yes

Reviewer #2: Yes

Reviewer #3: Yes

4. Have the authors made all data underlying the findings in their manuscript fully available?

Reviewer #1: Yes

Reviewer #2: Yes

Reviewer #3: Yes

5. Is the manuscript presented in an intelligible fashion and written in standard English?

Reviewer #1: Yes

Reviewer #2: Yes

Reviewer #3: Yes

6. Review Comments to the Author

Reviewer #1: All comments have been addressed. Therefore, this manuscript is suitable for publication in PLoS ONE.

Reviewer #2: I had only minor comments on the original submission and these have all been addressed in this revision.

Reviewer #3: The authors answered all the questions appropriately, no further comments or experiments should be performed in my opinion.

7. PLOS authors have the option to publish the peer review history of their article (what does this mean?). If published, this will include your full peer review and any attached files.

Reviewer #1: No

Reviewer #2: No

Reviewer #3: Yes: Lysangela R Alves

---

## [Editor Report · Acceptance letter]

18 Mar 2020

PONE-D-19-32465R1 

Biological functions of the autophagy-related proteins Atg4 and Atg8 in Cryptococcus neoformans 

Dear Dr. Vallim:

I am pleased to inform you that your manuscript has been deemed suitable for publication in PLOS ONE. Congratulations! Your manuscript is now with our production department. 

With kind regards,

on behalf of

Prof. Vladimir Trajkovic 

Academic Editor

PLOS ONE